# Cross-Domain Feature Alignment for Federated Domain Generalization

## Abstract

Learning a robust global model that generalizes well under domain skew is crucial for federated learning (FL). Feature alignment enhances domain-invariant representation learning, thereby aligning inconsistent feature spaces caused by domain skew. However, we find two key problems that limits feature alignment. (1) Mismatched batch normalization (BN) statistics and insufficient inter-class separation lead to divergent local prototypes under domain skew, preventing global prototypes from representing global information. (2) Existing feature alignment methods often introduce aggregation bias under domain skew, causing the feature space to favor domains with more clients. Building on these findings, we propose a novel federated learning approach with cross-domain feature alignment (FedCoda), which calibrates feature alignment and ensures fairness across domains. To learn domain-invariant features with feature alignment, FedCoda calibrates batch normalization and local prototypes to generate consistent representations across domains. To enhance the fairness of feature alignment across domains, FedCoda optimizes prototype aggregation and produces fair global prototypes. Extensive experiments show that FedCoda outperforms relevant baselines.

## 1 Introduction

Federated Learning (FL) (McMahan et al., 2017) is a distributed machine learning paradigm where clients collaborate to train a global model without sharing raw training data. One major challenge in FL is data heterogeneity, where the data distribution $P(\mathcal{X}, \mathcal{Y})$ varies across clients. By decomposing the joint distribution $P(\mathcal{X}, \mathcal{Y}) = P(\mathcal{X})P(\mathcal{Y}|\mathcal{X}) = P(\mathcal{Y})P(\mathcal{X}|\mathcal{Y})$, the data heterogeneity can be categorized into three distribution skews: (1) label distribution skew (Karimireddy et al., 2020; Li et al., 2020; Zhang et al., 2022), where label marginal distribution $P(\mathcal{Y})$ varies across clients; (2) feature distribution skew (Li et al., 2021; Huang et al., 2023), where feature marginal distribution $P(X)$ varies across clients; and (3) concept drift (Panchal et al., 2023; Chen et al., 2024a), where conditional distribution $P(\mathcal{Y}|\mathcal{X})$ varies across clients. Among these, feature distribution skew is particularly prevalent in real-world mobile and distributed environments. Data collected from heterogeneous sensing conditions, hardware platforms, or geographic regions can induce substantial domain-specific biases in representations. Such discrepancies yield inconsistent optimization directions among clients, degrading the generalization performance of the global model. This highlights the need for FL methods that explicitly address feature skew and promote the learning of domain-invariant representations.

Beyond discrepancies in source data distribution, a more practical challenge in FL is domain generalization (DG), which aims to train a model that generalizes well to unknown data distributions. Domain generalization has been extensively studied in centralized machine learning (Muandet et al., 2013; Motiian et al., 2017; Harary et al., 2022; Dayal et al., 2024), where raw data from multiple domains can be jointly utilized to learn domain-invariant representations. However, these techniques cannot be directly transferred to FL because they typically require centralized access to data across domains, which violates the privacy constraints of FL. This limitation motivates the central question:

*Q: How can we train a global model that generalizes well to unseen domains when raw data from different domains cannot be shared?*

To facilitate domain generalization under FL, GA (Zhang et al., 2023) and FedHEAL (Chen et al., 2024b) address this challenge by improving the fairness of model aggregation across heterogeneous clients. Orthogonal to these approaches, another promising direction is to enable domain-invariant representation learning under domain skew. To this end, feature alignment methods (Tan et al., 2022; Dai et al., 2023) have been proposed in FL, wherein local feature representations are aligned with a set of global prototypes. While effective in certain settings, these approaches overlook an important issue: *local prototypes may diverge significantly under domain skew.* Simply averaging these divergent prototypes to form global prototypes can induce unfair alignment, biasing the feature space toward domains with a larger number of clients.

To mitigate prototype bias, FPL (Huang et al., 2023) aggregates clustered domain prototypes and aligns local features with them. Although this strategy enhances fairness, it requires transmitting multiple clustered prototypes during each round, resulting in increased communication overhead and a risk of domain information leakage. Moreover, existing feature alignment methods generally rely on sharing category-wise statistics to estimate global prototypes, introducing additional privacy concerns. These limitations highlight the need for a feature alignment mechanism that supports domain generalization in FL while preserving communication efficiency and client privacy.

In addition to inducing unfair feature alignment, domain skew also amplifies bias in local prototypes. A key contributing factor is the use of Batch Normalization (BN) (Ioffe & Szegedy, 2015), whose statistics vary substantially across clients trained on domain-specific data. These discrepancies cause models with different BN statistics to produce inconsistent feature vectors, which in turn lead to divergent local prototypes. While prior studies have primarily examined how mismatched BN statistics hinder FL optimization and slow convergence (Wang et al., 2023; Zhang et al., 2024; Zhong et al., 2024; Yang et al., 2024), our work highlights their impact on feature alignment and prototype quality, an aspect that has received limited attention.

Beyond the BN mismatch issue, insufficient inter-class separation within each client further reduces prototype reliability. Under domain skew, features of different classes may become poorly separated, resulting in unstable or overlapping prototypes. When such divergent prototypes are aggregated, the resulting global prototypes fail to capture coherent global semantics and may distort the decision boundaries shared by clients. Consequently, global prototypes derived from these biased and weakly discriminative local prototypes cannot effectively represent global information, undermining the overall objective of domain-invariant representation learning.

To address the above challenges, we propose FedCoda, a cross-domain feature alignment method that calibrates local prototypes under domain skew and ensures fair alignment across domains. FedCoda comprises two core components: Calibrated Feature Alignment (CFA) and Fair Prototype Aggregation (FPA).

CFA enhances prototype consistency by employing global BN statistics during the later stages of training, thereby reducing representation discrepancies caused by domain-specific BN shifts. Our calibrated feature alignment (CFA) builds upon the two-stage BN calibration strategy introduced in FixBN (Zhong et al., 2024). However, directly applying FixBN in prototype-based federated domain generalization introduces additional challenges under domain skew, including unstable optimization and inconsistent prototype geometry. To address these issues, we extend FixBN with two key modifications: (i) an adaptive gradient clipping mechanism to stabilize training during BN calibration, and (ii) a prototype calibration loss that improves inter-class separation and prototype consistency. These extensions enable BN calibration to be effectively integrated with prototype-based feature alignment.

To ensure fairness across domains, FPA performs variance-minimizing prototype aggregation, which seeks global prototypes that minimize the dispersion of distances to local prototypes. This mitigates the dominance of domains with more clients. By aligning local representations with these unbiased prototypes, FedCoda promotes the learning of consistent and generalizable feature spaces, leading to a robust global model that generalizes well to unseen domains. In addition, FPA does not depend on local category distributions.

In summary, FedCoda tackles domain skew in FL through two complementary components. CFA mitigates inconsistencies in local representations caused by mismatched batch-normalization statistics and weak inter-class separation, producing reliable local prototypes. Building on these calibrated prototypes, FPA ensures unbiased global prototypes by preventing dominance from overrepresented domains. Together, these components enable robust domain-invariant representation learning and improved generalization under domain skew. Our main contributions are summarized as follows:

- We propose a calibrated feature alignment method that mitigates prototype discrepancies across clients and enhances inter-class separation, improving the reliability of local and global prototypes under domain skew (Sec. 3.3).

- We introduce a fairness-aware prototype aggregation method that minimizes cross-domain bias, enabling global prototypes to represent all domains more equitably (Sec. 3.4).

- We integrate the above components into FedCoda, a novel FL framework with cross-domain feature alignment that simultaneously improves consistency and fairness. Extensive experiments show that our FedCoda outperforms relevant baselines.

## 2 Related Work

### 2.1 Domain Generalization

Domain generalization (DG) aims to enhance generalizability to an unseen target domain by leveraging data from multiple source domains during training. Domain generalization has been extensively studied in centralized machine learning, with a number of approaches seeking to learn domain-invariant representations or domain-agnostic learning strategies. A line of work (Motiian et al., 2017; Li et al., 2018a;b; Zhao et al., 2020; Dayal et al., 2024) focuses on domain alignment, which minimizes the discrepancy among source domains in the feature space. These methods typically enforce similarity across domain-specific feature distributions, enabling the model to extract representations that generalize beyond the training domains. Another class of methods (Dou et al., 2019; Zhao et al., 2021; Balaji et al., 2018) adopts meta-learning techniques, treating each domain as a separate task. By simulating domain shifts during the meta-training process, these approaches encourage models to learn parameters that are robust to domain changes. Recent works (Harary et al., 2022; Bucci et al., 2021; Kim et al., 2021) have also explored connections between self-supervised learning and domain generalization, leveraging the invariances learned through auxiliary tasks to improve cross-domain generalization. Despite their effectiveness in centralized settings, these DG techniques rely on joint access to raw data from multiple domains, which is incompatible with the privacy constraints of federated learning. This limitation motivates the need for DG approaches specifically tailored to FL, where domain-invariant representations must be learned without direct data sharing.

### 2.2 Domain Skew in Federated Learning

Data heterogeneity poses a fundamental challenge in federated learning (FL), often leading to slow convergence and degraded global performance. Among the different forms of heterogeneity, domain skew, where feature distributions differ across clients, has received increasing attention due to its prevalence in real-world applications. To mitigate this issue, FedBN (Li et al., 2021) employs local batch normalization to isolate domain-specific statistics, reducing the negative impact of domain shifts. However, this personalization strategy prevents FedBN from generalizing effectively to unseen domains, limiting its applicability in domain-generalization settings. To improve generalization under domain skew, recent research has explored several complementary directions, including fair model aggregation (Chen et al., 2024b; Zhang et al., 2023; Tenison et al., 2022), refined optimization strategies (Nguyen et al., 2022; Qu et al., 2022; Guo et al., 2023; Wei & Han, 2024), and disentangled representation learning (Luo et al., 2022). A more recent line of work focuses on addressing inconsistent feature spaces across clients by aligning representations. Prototype-based feature alignment approaches (Ye et al., 2023; Huang et al., 2023; Zhu et al., 2023; Li et al., 2024; Le et al., 2025) aim to learn domain-invariant representations by aligning local features with aggregated global prototypes. FPL (Huang et al., 2023) enhances alignment fairness by clustering local prototypes and aggregating

unbiased prototypes, while I²PFL (Le et al., 2025) introduces a prototype reweighting scheme to balance the influence of clients during global prototype computation. Although these methods promote fairer feature alignment, they overlook a critical issue. Under domain skew, local prototypes themselves may be highly divergent due to inconsistent feature distributions and class separability. As a result, even fairly aggregated prototypes fail to represent global semantics effectively, limiting the overall capability of feature alignment methods.

### 2.3 BN in Federated Learning

Batch Normalization (BN) (Ioffe & Szegedy, 2015) is widely used to accelerate and stabilize the training of deep neural networks. However, in federated learning, BN becomes particularly fragile due to domain skew: clients trained on heterogeneous feature distributions accumulate inconsistent BN statistics, which propagate to the global model during aggregation and hinder convergence. This problem has motivated several attempts to replace BN with alternative normalization schemes. For example, (Hsieh et al., 2020) and (Zhang et al., 2024) employ Group Normalization (GN) (Wu & He, 2018) and Layer Normalization (LN) (Lei Ba et al., 2016), respectively, to avoid reliance on batch-level statistics. Nevertheless, (Chen & Chao, 2021) show that BN can still outperform GN in certain FL scenarios, suggesting that discarding BN entirely may not be optimal. To address the slow and unstable convergence caused by mismatched BN statistics, recent studies (Wang et al., 2023; Zhang et al., 2024; Zhong et al., 2024) have attempted to eliminate the detrimental impact of this mismatch. Distinct from these efforts, which primarily examine BN mismatch from an optimization perspective, this work analyzes its adverse impact on feature alignment, a critical yet underexplored aspect. We show that mismatched BN statistics produce inconsistent feature vectors, leading to significantly divergent local prototypes. This phenomenon undermines prototype-based alignment methods and limits the effectiveness of domain-invariant representation learning in federated learning.

## 3 Methodology

### 3.1 Problem Definition

We consider the standard federated learning (FL) setting consisting of a central server and $K$ clients. Each client $k$ holds a local dataset $D_k$ drawn from its own data distribution $P_k(\mathcal{X}, \mathcal{Y})$, where $\mathcal{X}$ and $\mathcal{Y}$ denote the input and label spaces, respectively. Let $\mathcal{L}$ be the loss function associated with a model $\boldsymbol{w}$ and a training sample $(\boldsymbol{x}, y)$. The local learning objective for client $k$ is defined as $F_k(\boldsymbol{w}) := \mathbb{E}_{(\boldsymbol{x}, y) \sim D_k} [\mathcal{L}_k(\boldsymbol{w}; \boldsymbol{x}, y)]$.

The FL task aims to learn a global model that minimizes the weighted average of local objectives. Formally, the global optimization problem can be formulated as:

$$\min_{\boldsymbol{w} \in \mathbb{R}^d} \left\{ F(\boldsymbol{w}) := \sum_{k=1}^{K} \frac{N_k}{N} F_k(\boldsymbol{w}) \right\}, \tag{1}$$

where $N_k = |D_k|$ is the size of each local dataset $D_k$ and $N = \sum_{k=1}^{K} N_k$ is the total number of training samples. At round $t$, the global model is updated by aggregating the local models as:

$$\boldsymbol{w}^{(t+1)} = \sum_{k=1}^{K} \frac{N_k}{N} \boldsymbol{w}_k^{(t)}, \tag{2}$$

where $\boldsymbol{w}_k^{(t)}$ represents the local model of client $k$ at round $t$.

In this work, we focus on the heterogeneity of feature distribution, also referred to as domain skew. Under domain skew, the marginal feature distribution $P(\mathcal{X})$ varies across clients, while the conditional distribution $P(\mathcal{Y}|\mathcal{X})$ remains consistent, i.e., $P_i(\mathcal{Y}|\mathcal{X}) = P_j(\mathcal{Y}|\mathcal{X})$ and $P_i(\mathcal{X}) \neq P_j(\mathcal{X})$. Furthermore, in federated domain generalization, the primary goal is to train a model that generalizes well to an unseen domain $\mathcal{D}_o$. Formally, the corresponding objective can be formulated as:

$$\min_{\boldsymbol{w} \in \mathbb{R}^d} \mathbb{E}_{(\boldsymbol{x}, y) \sim \mathcal{D}_o} [\mathcal{L}_k(\boldsymbol{w}; \boldsymbol{x}, y)]. \tag{3}$$

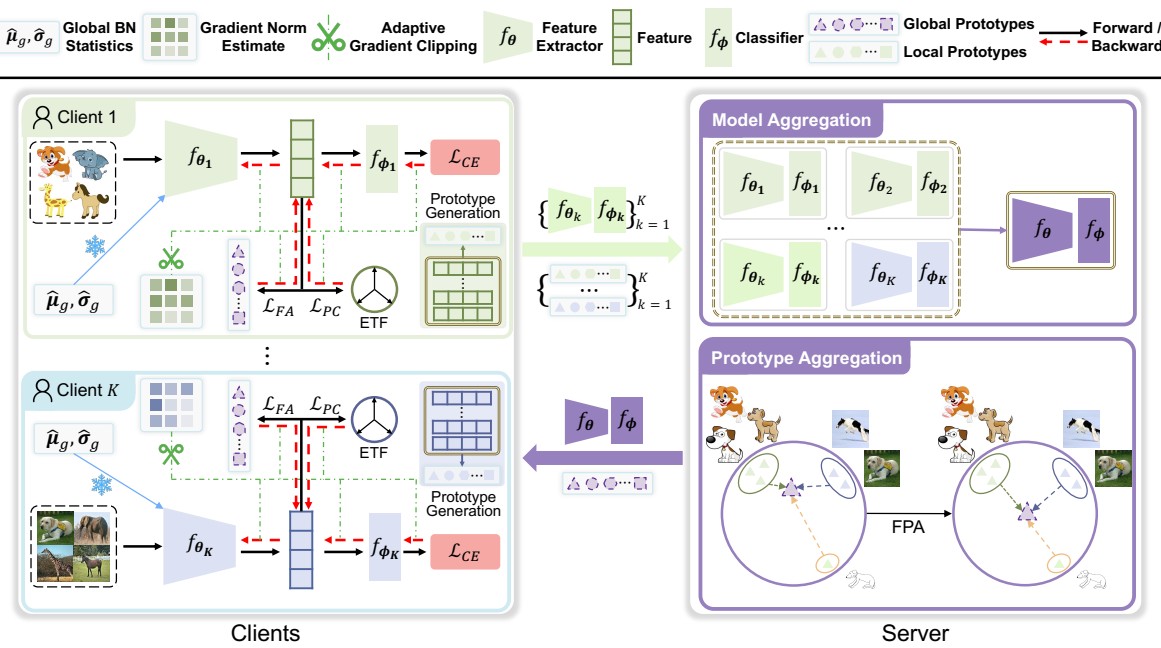

Figure 1: Overview of our cross-domain feature alignment (FedCoda). FedCoda involves four key steps: local training with global prototypes, local prototype generation, global model aggregation and global prototype aggregation. We denote different classes using distinct shapes and different domains using distinct colors.

Domain skew induces inconsistent feature spaces across clients. To mitigate feature inconsistency, feature alignment methods (Tan et al., 2022; Ye et al., 2023; Huang et al., 2023) aim to enforce domain-invariant representation learning through global prototypes. The global prototypes are aggregated from the local prototypes generated by each client. Specifically, for each client $k$, the model can be decoupled into a feature extractor $f_{\boldsymbol{\theta}_k}$ and a classifier $f_{\boldsymbol{\phi}_k}$. Given an input $\boldsymbol{x} \in \mathcal{X}$, the feature extractor $f_{\boldsymbol{\theta}_k} : \mathcal{X} \to \mathcal{Z}$ maps the input $\boldsymbol{x}$ into the feature space $\mathcal{Z}$ and generates a feature vector $\boldsymbol{z} = f_{\boldsymbol{\theta}_k}(\boldsymbol{x}) \in \mathcal{Z}$. Then, the classifier $f_{\boldsymbol{\phi}_k}$ maps the feature vector $\boldsymbol{z}$ into the class space $\mathbb{R}^C$. For each class $c$, the local prototype $\boldsymbol{p}_{k,c}$ is defined as the mean of the features corresponding to this class:

$$\boldsymbol{p}_{k,c} = \frac{1}{|D_k^c|} \sum_{(\boldsymbol{x},c) \in D_k^c} f_{\boldsymbol{\theta}_k}(\boldsymbol{x}), \quad \forall c \in [C]. \tag{4}$$

After local training, the server aggregates the local prototypes to form the global prototype $\bar{\boldsymbol{p}}_c$ for each class $c$:

$$\bar{\boldsymbol{p}}_c = \frac{1}{K} \sum_{k=1}^{K} \boldsymbol{p}_{k,c}, \quad \forall c \in [C]. \tag{5}$$

Note that the weighted aggregation $\bar{\boldsymbol{p}}_c = \sum_{k=1}^{K} \frac{N_{k,c}}{N_c} \boldsymbol{p}_{k,c}$ can leak privacy about class distribution, where $N_{k,c}$ denotes the number of samples of class $c$ in dataset $D_k$ and $N_c = \sum_{k=1}^{K} N_{k,c}$. Global prototypes are subsequently broadcast to clients, which align their feature spaces using prototype-based supervised objectives.

## 3.2 Motivation

Despite their effectiveness in reducing feature inconsistency, existing prototype alignment methods encounter two fundamental issues under domain skew: inconsistent local prototypes and unfair prototype aggregation. Local models trained on different domain data accumulate dissimilar BN statistics, resulting in divergent local prototypes. In addition to mismatched BN statistics, insufficient separation between class prototypes

leads to inconsistent decision boundaries. Consequently, global prototypes aggregated from these inconsistent local prototypes fail to effectively capture global information. Moreover, simple averaging biases the global prototypes toward domains with more clients. This results in unfair feature alignment, where local models from minority domains are encouraged to align with prototypes that poorly represent their underlying distributions. Further details on these two issues are provided in Sec. 3.3 and Sec. 3.4, respectively. These limitations motivate the design of FedCoda, which calibrates local prototype generation and enforces fairness during prototype aggregation. An overview of our FedCoda is presented in Figure 1 and the full procedure is summarized in Algorithm 1.

## 3.3 Calibrated Feature Alignment

Batch normalization (BN) is widely used to accelerate and stabilize deep neural network training. It uses the mean and variance of the mini-batch to normalize the feature map. Given a mini-batch $\mathcal{B}$ of $m$ features, a BN layer normalizes the features $\{\boldsymbol{x}_1, \boldsymbol{x}_2, \ldots, \boldsymbol{x}_m\}$ using the batch mean $\boldsymbol{\mu}_{\mathcal{B}}$ and variance $\boldsymbol{\sigma}_{\mathcal{B}}^2$, followed by a learnable affine transformation:

$$\hat{\boldsymbol{x}}_i = \boldsymbol{\gamma} \frac{\boldsymbol{x}_i - \boldsymbol{\mu}_{\mathcal{B}}}{\sqrt{\boldsymbol{\sigma}_{\mathcal{B}}^2 + \epsilon}} + \boldsymbol{\beta}, \tag{6}$$

where $\boldsymbol{\gamma}, \boldsymbol{\beta}$ are learnable parameters and $\epsilon$ ensures numerical stability. During local training, BN maintains moving averages of the mean $\boldsymbol{\mu}$ and variance $\boldsymbol{\sigma}$:

$$\hat{\boldsymbol{\mu}} \leftarrow \alpha \boldsymbol{\mu}_{\mathcal{B}} + (1 - \alpha)\hat{\boldsymbol{\mu}}, \tag{7}$$

$$\hat{\boldsymbol{\sigma}} \leftarrow \alpha \boldsymbol{\sigma}_{\mathcal{B}} + (1 - \alpha)\hat{\boldsymbol{\sigma}}. \tag{8}$$

Although BN is effective in centralized learning, recent studies (Wang et al., 2023; Zhong et al., 2024) have pointed out that mismatched BN statistics lead to deviated gradients across local models, hindering the convergence of the global model. Orthogonal to these works, we investigate *the adverse effects of mismatched BN statistics on feature alignment.* This issue has been largely overlooked by existing feature alignment methods.

The feature alignment method relies on global prototypes to unify the feature spaces across clients during local training. These global prototypes are aggregated from local prototypes generated by each client, as defined in Eq. (5). Local prototypes are obtained by averaging the representations of all local samples, as defined in Eq. (4). However, under domain skew, feature extractors $f_{\boldsymbol{\theta}}$ accumulate different BN statistics, resulting in inconsistent feature embeddings even for the same class, leading to divergent local prototypes. This divergence disturbs the generation of global prototypes and consequently misleads the feature alignment during local training. Specifically, due to the varying marginal distributions $P(\mathcal{X})$ across clients, the feature extractors trained on different domains accumulate different BN statistics $\{\hat{\boldsymbol{\mu}}, \hat{\boldsymbol{\sigma}}\}$. Assume that the feature extractor $f_{\boldsymbol{\theta}_k}$ contains $L$ layers and $\mathbf{A}^l(\boldsymbol{a}) = \mathbf{W}^l \boldsymbol{a} + \boldsymbol{b}^l$ is a linear transformation for the input $\boldsymbol{a}$, then the forward pass of feature extractor $f_{\boldsymbol{\theta}_k}$ can be formulated as:

$$\rho \circ f_{BN}^L \circ \mathbf{A}^L \circ \rho \circ f_{BN}^{L-1} \circ \mathbf{A}^{L-1} \circ \ldots \rho \circ f_{BN}^1 \circ \mathbf{A}^1, \tag{9}$$

where $\rho$ denotes the activation function. Since the marginal distribution $P(\mathcal{X})$ varies across clients, $\rho \circ f_{BN}^1 \circ \mathbf{A}_1$ also varies, and this divergence accumulates throughout the forward pass. This accumulated divergence causes each local model to produce divergent representations and thus divergent prototypes, as described in Eq. (9) and Eq. (4), respectively. Finally, the global prototypes aggregated from these divergent local prototypes fail to effectively represent global information, thereby hindering feature alignment.

### 3.3.1 BN Calibration under Domain Skew

To generate consistent local prototypes, we first calibrate BN behavior during local training. In the early rounds of FL, clients use their mini-batch BN statistics $\{\boldsymbol{\mu}_{\mathcal{B}}, \boldsymbol{\sigma}_{\mathcal{B}}\}$ to normalize the feature maps as Eq. (6). Then, all clients send their local BN statistics $\{\{\hat{\boldsymbol{\mu}}_k, \hat{\boldsymbol{\sigma}}_k\}\}_{k=1}^K$, and the server aggregates the global BN statistics $\{\hat{\boldsymbol{\mu}}_g, \hat{\boldsymbol{\sigma}}_g\}$ from local BN statistics. This stage follows the training process of FedAvg (McMahan et al., 2017). Once training enters the later stage, the global model tends to converge, and the BN statistics

of the global model can represent the overall distribution of all clients' local data. Although global BN statistics are less accurate than local BN statistics for training local model, they provide more generalizable and uniform normalization across clients. This helps train more consistent models across data from different domains and leads to more consistent local prototypes. Inspired by FixBN (Zhong et al., 2024), FedCoda switches to using global BN statistics during the later stage of FL. In practice, the calibration stage is implemented using a simple training schedule rather than requiring precise knowledge of the convergence point. We initialize training using local BN statistics and switch to global BN statistics after a certain number of communication rounds. In our experiments, we use the midpoint of the training process (i.e., $T/2$) as a practical default. Importantly, the effectiveness of this schedule does not depend on accurately identifying the exact convergence point. As shown in the sensitivity analysis in Sec. 4.4.3, the performance of FedCoda remains stable across a range of calibration rounds. This indicates that the calibration stage is not a sensitive hyperparameter but rather a coarse scheduling strategy that facilitates a smooth transition from domain-specific normalization to shared normalization across clients.

Although FixBN addresses the instability of batch normalization in federated learning, its objective differs from ours. FixBN focuses on improving optimization stability by fixing BN statistics after a certain training stage. In contrast, our study investigates the impact of BN mismatch on prototype-based feature alignment under domain skew. In prototype alignment methods, feature representations from different clients are explicitly aligned with shared global prototypes. When BN statistics differ significantly across clients, identical semantic classes may produce inconsistent feature embeddings, which leads to divergent local prototypes. These inconsistent prototypes subsequently distort global prototype aggregation and weaken feature alignment. Therefore, in our setting BN mismatch affects not only optimization but also the quality of learned representations, making prototype calibration particularly important. However, directly applying global BN statistics (i.e., vanilla FixBN) under domain skew can lead to gradient explosion, because the mismatch between local and global statistics is substantial. Therefore, vanilla FixBN is ill-suited for domain skew, and additional stabilization mechanisms are necessary.

### 3.3.2 Adaptive Gradient Clipping

To prevent gradient explosion, traditional gradient clipping strategy (Pascanu et al., 2013) rescales gradients whenever they go over a predefined norm threshold. This strategy needs to carefully turn a clipping threshold. However, in FL with domain skew, a fixed threshold is suboptimal. Specifically, different clients require different thresholds depending on the discrepancy between global BN statistics and their local BN statistics. Besides, as local models gradually adapt to global BN statistics, the threshold should increase to speed up model convergence. Considering these two limitations, we propose an adaptive gradient clipping strategy that automatically adjusts the clipping threshold per parameter. Unlike conventional clipping with a fixed threshold, which may either over-constrain updates or fail to prevent instability, our method estimates the clipping threshold using a smoothed history of gradient norms. This design allows each parameter to adapt its clipping scale according to the recent gradient statistics, providing stable optimization while preserving sufficient update magnitude for convergence. Once the current gradient norm exceeds this estimate by a significant margin, the gradient clipping mechanism is applied. Specifically, during local training, the gradient of $i$-th parameter at step $s$ is adjusted as follows:

$$\tilde{\boldsymbol{g}}_i^s \leftarrow \delta_i^s \cdot \boldsymbol{g}_i^s, \text{where } \delta_i^s = \min\left\{\frac{\alpha \cdot \gamma_i^{s-1}}{\|\boldsymbol{g_i^s}\|}, 1.0\right\}. \tag{10}$$

Here, $\gamma_i^s = \beta\gamma_i^{s-1} + (1-\beta)\|\boldsymbol{g}_i^s\|$ is the smoothed estimate of historical gradient norm for $i$-th parameter and $\beta$ is the decay factor. $\alpha$ is a hyperparameter to control the relative clipping threshold. Compared with fixed clipping, our method adapts threshold scaling through $\alpha \cdot \gamma_i^{s-1}$, thereby stabilizing training while accelerating the convergence of local models. We provide a theoretical analysis for the proposed adaptive gradient clipping mechanism used during the BN calibration stage. We make the following standard assumptions.

**Assumption 1 (Smoothness).** For each client $k$, the local objective $F_k(\boldsymbol{w})$ is $L$-smooth, i.e., for any $\boldsymbol{u}, \boldsymbol{v}$,

$$F_k(\boldsymbol{u}) \le F_k(\boldsymbol{v}) + \langle \nabla F_k(\boldsymbol{v}), \boldsymbol{u} - \boldsymbol{v} \rangle + \frac{L}{2}\|\boldsymbol{u} - \boldsymbol{v}\|^2. \tag{11}$$

**Assumption 2 (Unbiased stochastic gradients).** For each client $k$, the stochastic gradient estimator satisfies

$$\mathbb{E}[\boldsymbol{g}^s] = \nabla F_k(\boldsymbol{w}^s). \tag{12}$$

**Assumption 3 (Bounded gradient variance).** For each client $k$, the stochastic gradients satisfy

$$\mathbb{E}\big[\|\boldsymbol{g}^s - \nabla F_k(\boldsymbol{w}^s)\|^2\big] \le \sigma^2. \tag{13}$$

The following lemma characterizes the basic stability properties of the adaptive clipping rule.

**Lemma 1** (Norm control of adaptive clipping)**.** *For every parameter $i$ and iteration $s$, the clipped gradient satisfies*

$$\|\tilde{\boldsymbol{g}}_i^s\| \le \alpha \gamma_i^{s-1}. \tag{14}$$

*Furthermore, the deviation introduced by clipping is bounded by*

$$\|\boldsymbol{g}_i^s - \tilde{\boldsymbol{g}}_i^s\| \le \max\{\|\boldsymbol{g}_i^s\| - \alpha \gamma_i^{s-1}, 0\}. \tag{15}$$

**Proof.** If $\|\boldsymbol{g}_i^s\| \le \alpha \gamma_i^{s-1}$, then $\delta_i^s = 1$ and $\tilde{\boldsymbol{g}}_i^s = \boldsymbol{g}_i^s$, so the bound holds trivially. Otherwise,

$$\delta_i^s = \frac{\alpha \gamma_i^{s-1}}{\|\boldsymbol{g}_i^s\|},$$

which implies

$$\|\tilde{\boldsymbol{g}}_i^s\| = \delta_i^s \|\boldsymbol{g}_i^s\| = \alpha \gamma_i^{s-1}.$$

The bound in Eq. (15) follows immediately from the definition of $\tilde{\boldsymbol{g}}_i^s$. The bias bound follows directly from the definition of $\tilde{\boldsymbol{g}}_i^s$. $\qquad\square$

Lemma 1 shows that the update magnitude is controlled relative to the recent optimization history captured by $\gamma_i^{s-1}$. We next show that this control prevents the local update energy from becoming arbitrarily large.

**Theorem 1** (Stability of adaptive gradient clipping)**.** *Consider one local update on client $k$:*

$$\boldsymbol{w}^{s+1} = \boldsymbol{w}^s - \eta \tilde{\boldsymbol{g}}^s, \tag{16}$$

*where $\tilde{\boldsymbol{g}}^s$ is the clipped stochastic gradient obtained from Eq. (10). Under Assumption 1,*

$$F_k(\boldsymbol{w}^{s+1}) \le F_k(\boldsymbol{w}^s) - \eta \langle \nabla F_k(\boldsymbol{w}^s), \tilde{\boldsymbol{g}}^s \rangle + \frac{L\eta^2}{2} \|\tilde{\boldsymbol{g}}^s\|^2. \tag{17}$$

*Furthermore, by Lemma 1,*

$$\|\tilde{\boldsymbol{g}}^s\|^2 = \sum_i \|\tilde{\boldsymbol{g}}_i^s\|^2 \le \alpha^2 \sum_i (\gamma_i^{s-1})^2. \tag{18}$$

*Consequently,*

$$F_k(\boldsymbol{w}^{s+1}) \le F_k(\boldsymbol{w}^s) - \eta \langle \nabla F_k(\boldsymbol{w}^s), \tilde{\boldsymbol{g}}^s \rangle + \frac{L\eta^2 \alpha^2}{2} \sum_i (\gamma_i^{s-1})^2. \tag{19}$$

**Proof.** Applying the $L$-smoothness inequality in Assumption 1 with $\boldsymbol{u} = \boldsymbol{w}^{s+1}$ and $\boldsymbol{v} = \boldsymbol{w}^s$ gives

$$F_k(\boldsymbol{w}^{s+1}) \le F_k(\boldsymbol{w}^s) + \langle \nabla F_k(\boldsymbol{w}^s), \boldsymbol{w}^{s+1} - \boldsymbol{w}^s \rangle + \frac{L}{2} \|\boldsymbol{w}^{s+1} - \boldsymbol{w}^s\|^2.$$

Substituting $\boldsymbol{w}^{s+1} - \boldsymbol{w}^s = -\eta \tilde{\boldsymbol{g}}^s$ yields Eq. (17). Then Eq. (18) follows by summing Eq. (14) over all parameters. Plugging Eq. (18) into Eq. (17) gives Eq. (19). $\qquad\square$

The above theorem shows that adaptive clipping prevents the quadratic term in the smoothness-based descent bound from exploding. This property is particularly relevant during the BN calibration stage, where switching from local BN statistics to global BN statistics may induce abrupt changes in feature normalization and hence large gradient magnitudes under domain skew.

The next proposition further shows that the clipped stochastic gradient remains a controlled approximation of the true gradient.

**Proposition 1** (Expected deviation from the true gradient). *Under Assumptions 2 and 3, the clipped update satisfies*

$$\mathbb{E}\big[\|\tilde{\boldsymbol{g}}^s - \nabla F_k(\boldsymbol{w}^s)\|^2\big] \le 2\,\mathbb{E}\big[\|\tilde{\boldsymbol{g}}^s - \boldsymbol{g}^s\|^2\big] + 2\sigma^2. \tag{20}$$

**Proof.** By triangle inequality and $(\boldsymbol{a}+\boldsymbol{b})^2 \le 2\boldsymbol{a}^2 + 2\boldsymbol{b}^2$,

$$\|\tilde{\boldsymbol{g}}^s - \nabla F_k(\boldsymbol{w}^s)\|^2 \le 2\|\tilde{\boldsymbol{g}}^s - \boldsymbol{g}^s\|^2 + 2\|\boldsymbol{g}^s - \nabla F_k(\boldsymbol{w}^s)\|^2.$$

Taking expectation on both sides and using Assumption 3 proves the result. $\qquad\square$

Proposition 1 indicates that the deviation of the clipped update from the true gradient is controlled by two terms: the intrinsic stochastic gradient variance $\sigma^2$ and the clipping-induced bias $\|\tilde{\boldsymbol{g}}^s - \boldsymbol{g}^s\|^2$. By Lemma 1, this bias is nonzero only when the current gradient norm exceeds the historical scale $\alpha\gamma_i^{s-1}$. Therefore, the proposed mechanism mainly suppresses transient gradient spikes, which are common immediately after switching to global BN statistics under domain skew. Once training stabilizes, it behaves similarly to standard SGD.

### 3.3.3 Prototype Calibration under Domain Skew

Beyond BN mismatch, another source of prototype inconsistency arises from insufficient inter-class separation. Existing feature alignment methods (Ye et al., 2023; Huang et al., 2023) often employ contrastive objectives to push features away from prototypes of other classes. However, these approaches typically operate at the instance level and do not *explicitly enforce a structured geometry among class prototypes*. To promote stable and discriminative prototypes, we introduce a prototype calibration loss that encourages class prototypes to approximate a simplex equiangular tight frame (ETF) (Sustik et al., 2007). ETF structures have been shown to maximize angular separation among class representations, which improves class separability in representation learning.

Let $\bar{\boldsymbol{z}}_c$ denote the mean feature representation of class $c$ within a mini-batch:

$$\bar{\boldsymbol{z}}_c = \frac{1}{B_c}\sum_{i=1}^{B} \mathbf{1}_{\{y_i=c\}} \cdot \boldsymbol{z}_i, \tag{21}$$

and $\hat{\boldsymbol{z}}_c = \bar{\boldsymbol{z}}_c/\|\bar{\boldsymbol{z}}_c\|$ denote its normalized form. We encourage maximal angular separation among these normalized class representations by penalizing their pairwise similarities. The resulting loss is defined as:

$$\mathcal{L}_{PC} = \frac{1}{C}\sum_{i=1}^{C} \log\frac{1}{C-1}\sum_{j\ne i, j\in[C]} \exp(\hat{\boldsymbol{z}}_i^\top \hat{\boldsymbol{z}}_j). \tag{22}$$

If a class does not appear in the batch, the corresponding global prototype is used to compute $\mathcal{L}_{PC}$. This loss enforces maximal angular separation among classes, yielding more stable and consistent prototypes. Unlike standard contrastive learning losses that operate at the instance level, this objective operates directly on class prototypes and regularizes their geometric configuration. By encouraging an ETF-like structure among class prototypes, the proposed loss improves inter-class separation and leads to more stable prototype alignment across clients.

### 3.3.4 Calibrated Feature Alignment

The feature alignment loss for a sample $(\boldsymbol{x}, c)$ with label $c$ is defined as:

$$\mathcal{L}_{FA} = -\log\frac{\exp(sim(f_{\boldsymbol{\theta}}(\boldsymbol{x}), \bar{\boldsymbol{p}}_c)/\tau)}{\sum_{i=1}^{C}\exp(sim(f_{\boldsymbol{\theta}}(\boldsymbol{x}), \bar{\boldsymbol{p}}_i)/\tau)}, \tag{23}$$

where $\tau$ is a temperature parameter and $sim(f_{\boldsymbol{\theta}}(\boldsymbol{x}), \bar{\boldsymbol{p}}_i)$ denotes cosine similarity function. The feature alignment loss pulls features toward their corresponding global prototypes and pushes them away from

the global prototypes of the other classes. This will decrease intra-class variation and increase inter-class separation, contributing to more consistent feature spaces across clients. Finally, the total loss function during local training can be formulated as follows:

$$\mathcal{L} = \mathcal{L}_{CE} + \lambda_{FA}\mathcal{L}_{FA} + \lambda_{PC}\mathcal{L}_{PC}. \tag{24}$$

Here, $\mathcal{L}_{CE}$ is CrossEntropy loss, and $\lambda_{FA}$ and $\lambda_{PC}$ are the coefficients of feature alignment and prototype calibration, respectively.

### 3.4 Fair Prototype Aggregation

In federated domain generalization, most existing methods often overlook a challenging yet practical scenario in which some domains dominate the training process, i.e., a large fraction of clients share similar marginal feature distribution $P(\mathcal{X})$. Under this imbalance, global prototypes aggregated via simple averaging tend to be biased toward dominant domains. This can impair the global model's ability to generalize to unseen domains. To ensure that the global prototypes fairly represent all training domains, we propose a fair prototype aggregation (FPA) method that explicitly counteracts domain dominance. The key intuition behind FPA is that a fair global prototype should be equally representative of all domains. Concretely, this can be achieved by balancing the distances between each local prototype and the global prototype, preventing the latter from drifting toward any subset of domains.

To achieve this goal, we formulate fair prototype aggregation as an optimization problem. Specifically, for each class $c$ at communication round $t$, let $\{\boldsymbol{p}_{k,c}^{(t)}\}_{k=1}^{K}$ denote the local prototypes. We aim to compute a global prototype $\bar{\boldsymbol{p}}_c^{(t)}$ that minimizes the variance of distances between itself and the local prototypes. Formally, the global prototype is iteratively optimized to minimize the distance variance $\mathrm{Var}\,(\mathcal{M}_c)$ across all local prototypes. $\mathcal{M}_c = \{\|\boldsymbol{p}_{k,c}^{(t)} - \bar{\boldsymbol{p}}_c^{(t)}\|_2\}_{k=1}^{K}$ denotes the Euclidean distances between each local prototype $\{\boldsymbol{p}_{k,c}^{(t)}\}_{k=1}^{K}$ and the global prototype $\bar{\boldsymbol{p}}_c^{(t)}$. We will refer to $\|\boldsymbol{p}_{k,c}^{(t)} - \bar{\boldsymbol{p}}_c^{(t)}\|_2$ as $d_{k,c}^{(t)}$ in the rest of paper.

Rather than directly optimizing from a potentially biased initialization, we first compute an initial point $\tilde{\boldsymbol{p}}_c^{(t)}$. While the arithmetic mean of local prototypes is a natural choice (i.e., the global prototype $\bar{\boldsymbol{p}}_c^{(t)}$ aggregated as Eq. (5)), it is often biased toward dominant domains and slows optimization. To mitigate this, we initialize the global prototype by assigning larger weights to local local prototypes that deviate more from the prototype mean, thereby compensating for underrepresented domains:

$$\tilde{\boldsymbol{p}}_c^{(t)} = \sum_{k=1}^{K} \frac{d_{k,c}^{(t)}}{\sum_{i=1}^{K} d_{i,c}^{(t)}} \boldsymbol{p}_{k,c}^{(t)}. \tag{25}$$

Starting from $\tilde{\boldsymbol{p}}_c^{(t)}$, we then optimize the global prototype by minimizing the variance of distances:

$$\arg\min_{\bar{\boldsymbol{p}}_c^{(t)}} \left\{ \frac{1}{K} \sum_{k=1}^{K} \left( d_{k,c}^{(t)} - \frac{1}{K} \sum_{i=1}^{K} d_{i,c}^{(t)} \right)^2 \right\}. \tag{26}$$

This objective encourages the global prototype to equidistant from all local prototypes, effectively neutralizing the influence of dominant domains. Under domain skew, local prototypes arise from heterogeneous feature distributions and typically form asymmetric clusters across domains. Therefore, this objective can produce a reasonable solution that represents the global information. We solve the above optimization using L-BFGS-B algorithm (Zhu et al., 1997), and the resulting solution is used as the updated global prototype $\bar{\boldsymbol{p}}_c^{(t+1)}$.

Unlike weighted prototype aggregation schemes that rely on category distributions, FPA operates solely on prototype geometry and does not rely on sharing explicit label distribution statistics. As a result, it ensures balanced representation across domains. Besides, FPA only adds transmission of class-wise prototypes, while FPL requires transmitting multiple clustered prototypes. When combined with calibrated feature alignment, FPA enables fair and robust domain-invariant representation learning in federated learning.

---

**Algorithm 1** FedCoda

---

1: **Input:** number of communication rounds $T$, initial model $\boldsymbol{w}$, number of local epochs $E$, learning rate $\eta$, prototype calibration coefficient $\lambda_{PC}$ and feature alignment coefficient $\lambda_{FA}$.
2: **for** $t = 0, 1, \ldots, T-1$ **do**
3:    # Server executes:
4:    Send global model $\boldsymbol{w}^{(t)}$ and global prototypes $\{\bar{\boldsymbol{p}}_c^{(t)}\}_{c \in [C]}$ to each client
5:    # Clients execute:
6:    **for** each client $k \in [K]$ in parallel **do**
7:       Set $\boldsymbol{w}_k^{(t)} = \boldsymbol{w}^{(t)}$
8:       **if** $t >= T/2$ **then**
9:          Switch to global BN statistics
10:       **else**
11:          Use local BN statistics
12:       **end if**
13:       **for** each training step $s$ **do**
14:          Sample a mini-batch $\mathcal{B}$
15:          Compute CrossEntropy loss $\mathcal{L}_{CE}$, and compute $\mathcal{L}_{PC}$ and $\mathcal{L}_{FA}$ by Eq. (22) and Eq. (23)
16:          $\mathcal{L} = \mathcal{L}_{CE} + \lambda_{FA}\mathcal{L}_{FA} + \lambda_{PC}\mathcal{L}_{PC}$
17:          $\boldsymbol{g}^s = \nabla_{\boldsymbol{w}_k^{(t)}} \mathcal{L}(\boldsymbol{w}_k^{(t)}, \mathcal{B})$
18:          **if** $t >= T/2$ **then**
19:             **for** $i \in |\boldsymbol{w}_k^{(t)}|$ **do**
20:                Clip $\boldsymbol{g}_i^s$ as Eq. (10)
21:                $\gamma_i^s = \beta\gamma_i^{s-1} + (1-\beta)\|\boldsymbol{g}_i^s\|$
22:             **end for**
23:          **end if**
24:          Update $\boldsymbol{w}_k^{(t)}$ by clipped $\boldsymbol{g}^s$
25:       **end for**
26:       **for** $c \in [C]$ **do**
27:          Generate local prototype $\boldsymbol{p}_{k,c}$ by Eq. (4)
28:       **end for**
29:       Send $\boldsymbol{w}_k^{(t)}$ and $\{\boldsymbol{p}_{k,c}\}_{c \in [C]}$ to server
30:    **end for**
31:    # Server executes:
32:    Update global model $\boldsymbol{w}^{(t+1)}$ by Eq. (2)
33:    **for** each class $c \in [C]$ **do**
34:       Compute prototype distances $\mathcal{M}_c$ and the initial point $\tilde{\boldsymbol{p}}_c^{(t)}$
35:       Update global prototype $\bar{\boldsymbol{p}}_c^{(t+1)}$ by Eq. (26)
36:    **end for**
37: **end for**

---

### 3.5 FedCoda

FedCoda addresses domain skew in federated learning through two complementary components. Calibrated Feature Alignment (CFA) focuses on improving the quality and consistency of local prototypes by mitigating the adverse effects of mismatched BN statistics and enhancing inter-class separation. By stabilizing representation learning at the client level, CFA enables the generation of reliable local prototypes. Building upon these calibrated prototypes, Fair Prototype Aggregation (FPA) ensures that the global prototypes are unbiased and domain-fair by preventing dominance from overrepresented domains. Together, CFA and FPA form a cohesive framework that produces consistent, discriminative, and fair global prototypes, enabling robust domain-invariant representation learning and improved generalization under domain skew.

The procedure of our FedCoda is formally presented in Algorithm 1. FedCoda involves four key steps: local training with global prototypes (Lines 7-25), local prototype generation (Lines 26-28), global model

aggregation (Line 32) and global prototype aggregation (Lines 33-36). During local training, global BN statistics is used (Lines 9-11) when reaching the calibration stage (i.e., the second half of the total FL training rounds). Note that feature alignment is not enabled in the first round of FL training (i.e., round 0), as global prototypes are not yet available during that round.

# 4 Experiments

## 4.1 Experimental Setup

### 4.1.1 Datasets

We evaluate the proposed method on three widely used domain generalization benchmarks: Digits (Zhou et al., 2020), PACS (Li et al., 2017) and DomainNet (Peng et al., 2019). For each dataset, we split the dataset into 80% for training and 20% for testing. The Digits dataset consists of four domains, namely MNIST (M), MNIST-M (MM), SVHN (SV) and SYN (SY). Each domain contains ten categories corresponding to digits from 0 to 9. The PACS dataset includes four domains, namely Photo (P), Art Painting (A), Cartoon (C) and Sketch (S). Each domain includes 7 categories (dog, elephant, giraffe, guitar, horse, house and person). DomainNet is a large-scale dataset comprising images from six distinct domains, including Clipart (C), Infograph (I), Painting (P), Quickdraw (Q), Real (R) and Sketch (S). Each domain contains 345 categories. Following FedBN (Li et al., 2021) and GA (Zhang et al., 2023), we construct a reduced version of DomainNet by selecting the ten most frequent categories to form a sub-dataset for our experiments. Representative samples from the three datasets are illustrated in Figure 2.

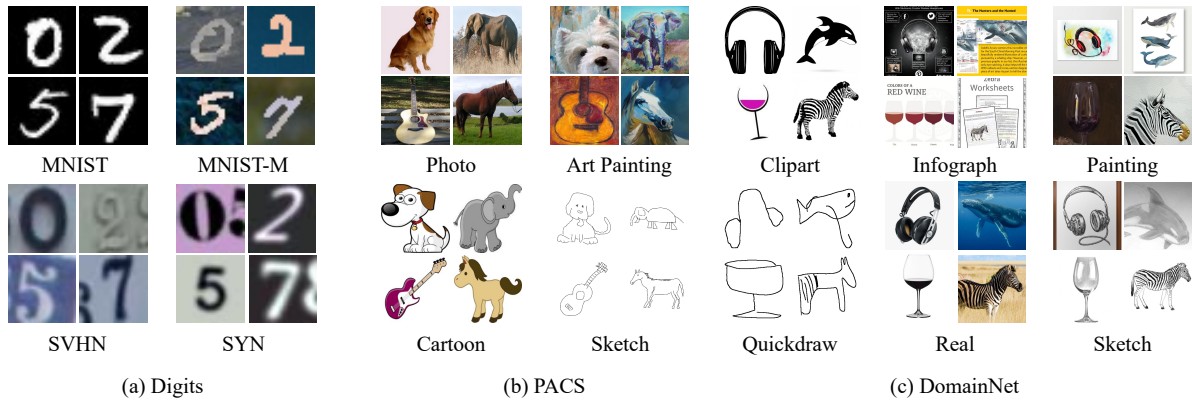

| MNIST | MNIST-M | Photo | Art Painting | Clipart | Infograph | Painting |
| SVHN | SYN | Cartoon | Sketch | Quickdraw | Real | Sketch |
| (a) Digits | | (b) PACS | | (c) DomainNet | | |

Figure 2: The representative samples from the three datasets.

### 4.1.2 Baselines

We compare FedCoda with representative methods from four categories: (1) standard federated learning method: FedAvg (McMahan et al., 2017); (2) federated learning methods with feature alignment: FedFM (Ye et al., 2023) and FPL (Huang et al., 2023); (3) federated learning methods under domain skew: FedHEAL (Chen et al., 2024b), GA (Zhang et al., 2023), FedSR (Nguyen et al., 2022), FedGMA (Tenison et al., 2022), and FedIIR (Guo et al., 2023); (4) federated learning method with fixed BN statistics: FixBN (Zhong et al., 2024). We note that the original FixBN method does not employ gradient clipping, which can lead to gradient explosion under domain skew (as shown in Table 6). *For a fair comparison, we augment FixBN with stardard gradient clipping using a fixed norm threshold of 10 in all experiments. This means that the reported FixBN includes additional clipping not present in the original method. Note that this addition does not change the core FixBN mechanism.*

### 4.1.3   Metrics

Following prior work (Li et al., 2021; Zhang et al., 2023; Guo et al., 2023), we adopt the "*leave-one-domain-out*" strategy for all datasets. Specifically, one domain is held out as the unseen test domain, while the remaining domains are used for training. The samples in each training domain are split into 80% for training and 20% for testing. To specifically evaluate the effectiveness of feature alignment, we do not employ ImageNet-pretrained ResNet18. Consequently, our results are not directly comparable to those reported in GA (Zhang et al., 2023) and FedIIR (Guo et al., 2023), which rely on pretrained backbones. Besides, a recent benchmark (Bai et al., 2024) uses ImageNet-pretrained ResNet50, which may show higher accuracy on some datasets. All experiments are repeated three times using different random seeds {1024, 2048, 4096}, and we report the average accuracy over the last 10 communication rounds to reduce stochasticity in FL training.

### 4.1.4   Client Distribution

To evaluate FedCoda and competing baselines under a realistic scenario, we consider heterogeneous client distributions across domains, where the number of clients varies across domains. Specifically, following FPL (Huang et al., 2023), we initialize 10 clients for Digits and PACS, and 12 clients for DomainNet. The client distributions are as follows: Digits is { M: 3, MM: 2, SV: 1, SY: 4 }, PACS is { P: 3, A: 2, C: 1, S: 4 }, and DomainNet is { C: 1, I: 1, P: 2, Q: 2, R: 3, S: 3 }. The key difference between our setting and that of FPL is that we adopt the "leave-one-domain-out" strategy. Within each domain, samples from each category are evenly partitioned among the corresponding clients, ensuring that clients belonging to the same domain share an identical marginal label distribution $P(\mathcal{Y})$. However, since $P(\mathcal{Y})$ may differ across domains, the marginal label distributions of local datasets can still vary across clients. Notably, for PACS and DomainNet, the number of samples per category differs substantially across domains. This results in the coexistence of domain skew (i.e., variations in $P(\mathcal{X})$) and label skew (i.e., variations in $P(\mathcal{Y})$), making the experimental setting particularly challenging and representative of real-world federated learning scenarios.

### 4.1.5   Implementation Details

All experiments are conducted for 100 communication rounds, with local epochs $E = 5$ per round. Clients perform local updates using SGD optimizer with a learning rate of 0.01. The weight decay is 0.00001 and the SGD momentum is 0.9. For all feature alignment methods, the alignment coefficient is set to 1.0 and the temperature $\tau$ is set to 0.1. For FedHEAL, the momentum of weight update is set to 0.5, with a threshold parameter $\tau = 0.5$. For GA, the step size $d$ is set to 0.05. For FedIIR, the penalty coefficient $\gamma$ and exponential moving average parameter $v$ are set to 0.001 and 0.95, respectively. For FedSR, $\alpha^{L2R}$ and $\alpha^{CMI}$ are set to 0.01 and 0.001, respectively. For FedGMA, $\tau$ is set to 0.4. For both FedCoda and FixBN, the BN calibration stage begins at round 50. For FedCoda, the prototype calibration weight $\lambda_{PC}$ is set to 1.0, and the adaptive gradient clipping parameters $\alpha$ and $\beta$ are set to 1.5 and 0.99, respectively. We employ ResNet10 for Digits and use ResNet18 (He et al., 2016) for PACS and DomainNet. Notably, unlike recent methods (Zhang et al., 2023; Guo et al., 2023), we do not use ImageNet-pretrained ResNet18. This design choice allows us to directly evaluate the effectiveness of representation learning induced by feature alignment.

## 4.2   Main Results

Table 1 and Table 2 report the Top-1 accuracy on Digits, PACS, and DomainNet under the leave-one-domain-out protocol, with results averaged over three runs. Overall, FedCoda consistently achieves the best performance across all three benchmarks, both on individual target domains and in terms of average accuracy, demonstrating its effectiveness for federated domain generalization under domain skew.

On Digits, FedCoda outperforms all baselines on every target domain, achieving an average accuracy of 81.1%. Similar trends are observed on PACS, where FedCoda achieves an average accuracy of 68.3%, outperforming FixBN by 3.1% and substantially surpassing standard FL and fair aggregation methods. On the more challenging DomainNet benchmark, FedCoda attains the highest average accuracy of 68.0%, which improves upon the strongest baseline (FixBN) by 1.3%. These results highlight the robustness of FedCoda in large-scale, multi-domain settings with severe domain imbalance.

Table 1: Top-1 accuracy (%) on Digits and PACS. All results are averaged over 3 runs (mean±std). The best results are highlighted in bold, while the second-best are underlined.

| Method | Digits | | | | | PACS | | | | |
|--------|--------|--------|--------|--------|--------|--------|--------------|--------|--------|--------|
| | MNIST | MNIST-M | SVHN | SYN | Avg. | Photo | Art Painting | Cartoon | Sketch | Avg. |
| FedAvg | 94.1±0.2 | 54.9±0.4 | 64.7±0.7 | 92.9±0.2 | 76.6±0.1 | 66.3±1.0 | 47.2±0.9 | 61.3±0.6 | 69.9±0.7 | 61.2±0.4 |
| FedFM | 94.8±0.3 | 55.7±0.3 | 67.2±0.7 | 93.8±0.1 | 77.9±0.2 | 71.7±0.5 | 51.6±0.9 | 63.4±0.8 | 73.0±0.3 | 64.9±0.3 |
| FPL | 93.6±0.5 | 55.6±0.3 | 66.8±0.7 | 93.9±0.1 | 77.4±0.3 | 72.1±0.4 | 51.6±1.0 | 63.1±0.7 | 72.3±0.5 | 64.8±0.3 |
| FedHEAL | 94.1±0.4 | 52.7±0.4 | 62.7±0.6 | 92.0±0.1 | 75.4±0.2 | 66.3±0.6 | 48.3±0.8 | 60.6±0.3 | 67.8±0.5 | 60.7±0.3 |
| FedGMA | 90.9±0.4 | 54.3±0.6 | 61.1±0.9 | 92.4±0.3 | 74.7±0.3 | 61.6±2.4 | 45.7±1.3 | 56.1±0.9 | 66.0±1.2 | 57.4±0.6 |
| GA | 95.0±0.2 | 53.0±0.4 | 65.1±0.4 | 93.3±0.1 | 76.6±0.1 | 65.0±0.9 | 51.9±0.9 | 56.5±0.8 | 67.4±0.7 | 60.2±0.2 |
| FedIIR | 94.8±0.3 | 54.7±0.4 | 64.8±0.7 | 93.4±0.2 | 76.9±0.2 | 66.1±0.8 | 47.6±0.8 | 61.6±0.7 | 69.8±1.2 | 61.2±0.5 |
| FedSR | 93.4±0.5 | 56.6±0.7 | 66.3±0.8 | 94.0±0.1 | 77.6±0.3 | 69.7±0.9 | 51.2±1.4 | 64.0±1.0 | 72.1±0.6 | 64.3±0.6 |
| FixBN | 95.8±0.1 | 59.7±0.5 | 72.3±0.6 | 93.9±0.1 | 80.4±0.2 | 73.0±0.6 | 51.4±0.9 | 65.4±0.6 | 71.2±0.4 | 65.2±0.4 |
| FedCoda | **96.5±0.2** | **61.2±0.3** | **72.6±0.6** | **94.2±0.1** | **81.1±0.2** | **76.8±0.2** | **57.3±0.5** | **66.0±0.5** | **73.3±0.3** | **68.3±0.2** |

Table 2: Top-1 accuracy (%) on DomainNet. All results are averaged over 3 runs (mean±std). The best results are highlighted in bold, while the second-best are underlined.

| Method | Clipart | Infograph | Painting | Quickdraw | Real | Sketch | Avg. |
|--------|---------|-----------|----------|-----------|------|--------|------|
| FedAvg | 76.3±0.4 | 35.7±0.2 | 68.8±0.4 | 53.5±1.1 | 70.3±0.5 | 79.9±0.4 | 64.1±0.2 |
| FedFM | 80.1±0.3 | 37.3±0.3 | 72.6±0.5 | 55.3±0.3 | 71.0±0.3 | 83.4±0.2 | 66.6±0.2 |
| FPL | 79.7±0.2 | 37.2±0.2 | 72.0±0.3 | 54.8±0.6 | 70.8±0.5 | 82.1±0.4 | 66.1±0.2 |
| FedHEAL | 76.8±0.5 | 35.8±0.2 | 69.5±0.4 | 53.0±0.6 | 69.6±0.7 | 80.3±0.4 | 64.2±0.1 |
| FedGMA | 74.5±0.7 | 34.2±0.7 | 67.9±1.9 | 48.8±1.7 | 66.9±1.8 | 78.0±1.4 | 61.7±0.4 |
| GA | 73.5±0.7 | 32.9±0.3 | 64.1±0.9 | 52.4±0.5 | 69.5±0.6 | 77.1±0.3 | 61.6±0.2 |
| FedIIR | 76.4±0.5 | 35.7±0.3 | 68.5±0.4 | 51.9±1.1 | 69.8±0.6 | 79.6±0.4 | 63.6±0.2 |
| FedSR | 78.2±2.0 | 37.5±0.2 | 72.2±0.5 | 53.2±1.3 | 71.7±1.2 | 81.8±0.8 | 65.8±0.6 |
| FixBN | 79.6±0.3 | 36.2±0.1 | 72.2±0.4 | 56.9±0.5 | 72.6±0.1 | 82.5±0.3 | 66.7±0.2 |
| FedCoda | **80.6±0.2** | **37.6±0.2** | **73.8±0.3** | **58.6±0.7** | **73.9±0.5** | **83.5±0.4** | **68.0±0.1** |

Among baselines, methods that explicitly incorporate feature alignment (e.g., FedFM and FPL) generally outperform standard FedAvg, confirming the importance of learning domain-invariant representations for domain generalization. However, their performance gains are limited when local prototypes are inconsistent or when global prototypes are biased toward dominant domains. Methods such as GA and FedIIR, which rely on generalization-aware aggregation or gradient alignment, perform less favorably in our setting. This is primarily because these methods assume well-pretrained global models. Without ImageNet pretraining, their validation signals or gradient alignment become unreliable, leading to inferior generalization to unseen domains.

In contrast, FedCoda addresses these limitations through two complementary mechanisms. First, Calibrated Feature Alignment (CFA) mitigates the adverse effects of mismatched BN statistics and enhances inter-class separation, producing more consistent and discriminative local prototypes. Second, Fair Prototype Aggregation (FPA) prevents global prototypes from being dominated by overrepresented domains, enabling balanced feature alignment across clients. Together, these components lead to more reliable global prototypes and improved decision boundaries, which translate into superior generalization performance across diverse unseen domains.

## 4.3 Ablation Study

### 4.3.1 Effects of Core Components

To assess the effectiveness of two core components in FedCoda, we conduct ablation studies on Digits and PACS, as reported in Table 3. The results show that both components contribute positively, and their combination yields the largest performance gains across all evaluated settings. On Digits, where domain shifts are relatively mild and visual styles are more homogeneous, local prototypes across clients tend to be similar.

Consequently, global prototypes aggregated via simple averaging are less biased, and the marginal benefit of FPA is limited, yielding an improvement of 1.2% when applied alone. In contrast, CFA provides a substantial gain, indicating that BN calibration and enhanced inter-class separation play a dominant role in stabilizing representation learning under moderate domain skew. On the more challenging PACS dataset, where domains differ significantly in style and data distribution, the effect of FPA becomes markedly more pronounced. Mean-based prototype aggregation leads to biased global prototypes dominated by overrepresented domains. When both components are enabled, FedCoda achieves the largest gain, highlighting the complementary nature of CFA and FPA under severe domain skew. Overall, CFA improves prototype quality, while FPA ensures fair aggregation, and both are necessary for robust federated domain generalization.

Table 3: Ablation study of core components of FedCoda on Digits and PACS. All results are averaged over 3 runs.

| FPA | CFA | Digits | | | | | | PACS | | | | | |
|-----|-----|------|------|------|------|------|------|------|------|------|------|------|------|
| | | M | MM | SV | SY | Avg. | Δ | P | A | C | S | Avg. | Δ |
| | | 94.1 | 54.9 | 64.7 | 92.9 | 76.6 | - | 66.3 | 47.2 | 61.3 | 69.9 | 61.2 | - |
| ✓ | | 94.2 | 55.4 | 67.9 | 93.7 | 77.8 | 1.2 | 71.2 | 51.5 | 63.6 | 72.6 | 64.7 | 3.5 |
| | ✓ | 96.3 | 60.9 | 72.8 | 94.0 | 80.9 | 4.3 | 75.8 | 57.0 | 65.4 | 72.5 | 67.6 | 6.4 |
| ✓ | ✓ | 96.5 | 61.2 | 72.6 | 94.2 | **81.1** | **4.5** | 76.8 | 57.3 | 66.0 | 73.3 | **68.3** | **7.1** |

Table 4: Ablation study of BN Calibration (BNC) and Prototype Calibration (PC) on Digits and PACS. All results are averaged over 3 runs.

| Dataset | Digits | | | | | PACS | | | | |
|---------|------|------|------|------|------|------|------|------|------|------|
| Accuracy | M | MM | SV | SY | Avg. | P | A | C | S | Avg. |
| FedCoda w/o BNC | 94.7 | 57.9 | 68.8 | 93.7 | 78.8 | 72.7 | 53.2 | 64.1 | 72.9 | 65.7 |
| FedCoda w/o PC | 96.4 | 59.6 | 72.1 | 93.7 | 80.5 | 76.2 | 56.2 | 64.9 | 72.2 | 67.4 |
| FedCoda | 96.5 | 61.2 | 72.6 | 94.2 | **81.1** | 76.8 | 57.3 | 66.0 | 73.3 | **68.3** |

We further examine the impact of BN Calibration (BNC) and Prototype Calibration (PC), two key components of CFA. We conduct an ablation study on Digits and PACS. Table 4 shows that BNC and PC consistently improve generalization performance on unseen domains, and BNC plays a more essential role.

### 4.3.2 Geometric Structure of Prototype Calibration

To better understand the geometric structure of prototype calibration, we visualize the learned feature spaces using t-SNE on PACS (Figure 3). Without PC, class clusters exhibit noticeable overlap, resulting in ambiguous decision boundaries. In contrast, enabling PC leads to clearer class separation and more compact intra-class clusters, indicating improved discriminability. To complement the qualitative analysis, we further evaluate representation quality using Normalized Mutual Information (NMI). Specifically, we apply K-means clustering to the learned features and compute NMI between cluster assignments and ground-truth labels. Table 5 shows that PC consistently increases NMI across all unseen domains, demonstrating that PC effectively enforces maximal inter-class separation by penalizing similarity among class prototypes. As a result, CFA not only stabilizes representations through BN calibration but also improves the geometric structure of the feature space, which directly translates into better generalization on unseen domains.

### 4.3.3 Effects of Gradient Clipping

As shown in Table 6, we conduct an ablation study to examine the role of gradient clipping. In the absence of gradient clipping, FixBN fails to converge when directly applying global BN statistics under domain skew. Incorporating gradient clipping significantly improves the performance of the global model. Moreover, our adaptive gradient clipping further enhances stability and yields additional performance gains.

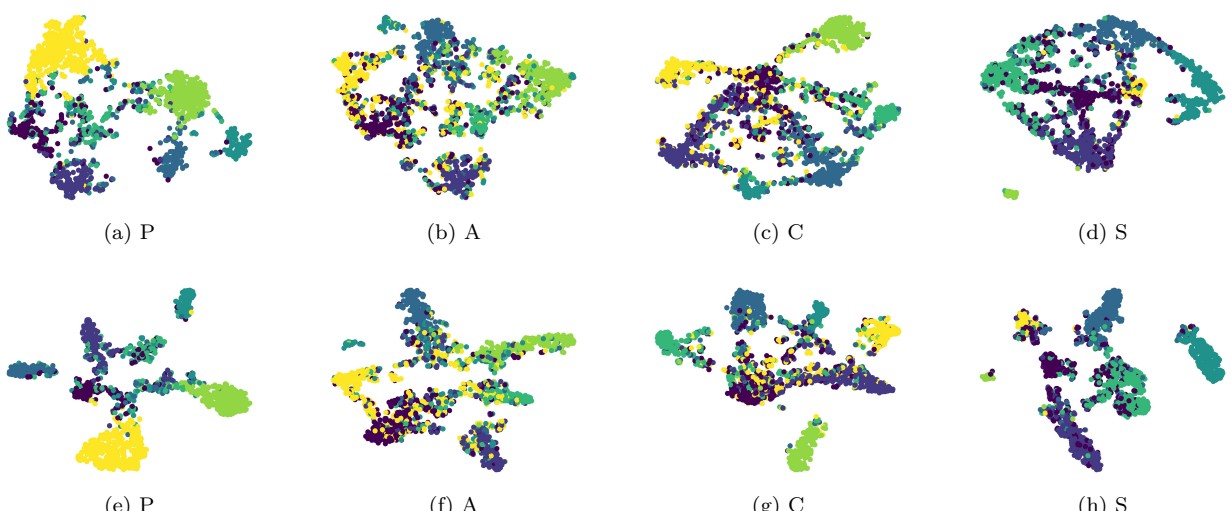

Figure 3: t-SNE visualization of representations on PACS dataset with four unseen domains. Top row: FedCoda without PC; bottom row: FedCoda with PC.

Table 5: NMI values of the feature spaces in Figure 3. Higher NMI values indicate better semantic consistency between the learned embedding space and the ground-truth labels.

| Methods | PACS | | | |
|---|---|---|---|---|
| | P | A | C | S |
| FedCoda w/o PC | 0.5342 | 0.2882 | 0.4333 | 0.4589 |
| FedCoda | **0.6452** | **0.3464** | **0.4450** | **0.4966** |

To better understand this behavior, we visualize the gradient norms over communication rounds. As shown in Figure 4, the gradient norms exhibit a sharp increase at the BN switching round (i.e., round 50), indicating a sudden change in optimization dynamics. Applying gradient clipping effectively controls these large gradients and stabilizes training during this transition.

Table 6: Top-1 accuracy on PACS with different gradient clipping strategies.

| Methods | PACS | | | |
|---|---|---|---|---|
| | P | A | C | S |
| FixBN w/o gradient clipping | 11.3±0.0 | 18.5±0.0 | 16.6±0.0 | 19.6±0.0 |
| FixBN with fixed clipping | 73.0±0.6 | 51.4±0.9 | 65.4±0.6 | 71.2±0.4 |
| FixBN with adaptive clipping | 74.7±0.5 | 53.8±0.9 | 64.5±0.6 | 71.7±0.4 |
| FedCoda | **76.8±0.2** | **57.3±0.5** | **66.0±0.5** | **73.3±0.3** |

### 4.3.4 Analysis of BN Mismatch

To evaluate the effects of BN mismatch on the consistency of prototypes, we measure the cosine similarity of local prototypes across clients. As shown in Table 7, BN calibration can enhance the prototype consistency across clients. In particular, this mismatch is more severe when the training domains contain Sketch (i.e., when the unseen domains are Photo, Art Painting and Cartoon), whose style is much different from other domains. However, BN calibration can effectively mitigate the mismatched BN statistics under this setting, thereby improving the consistency of local prototypes.

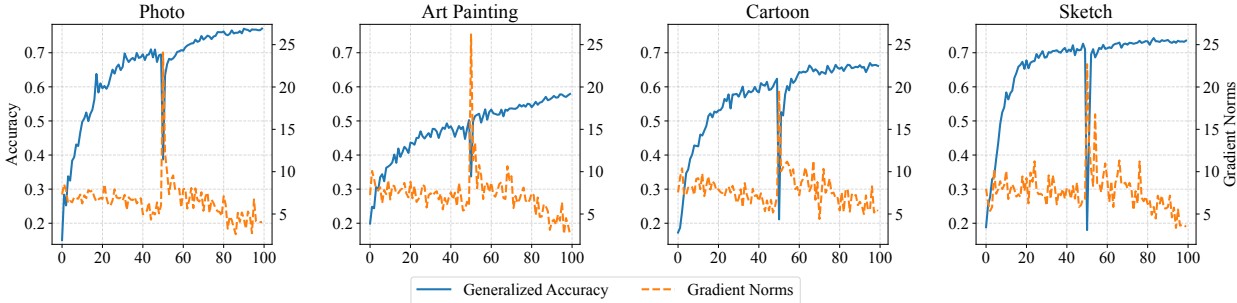

Figure 4: Generalized accuracy and gradient norms over communication rounds.

Table 7: Cosine similarity of local prototypes across clients.

| Methods | PACS | | | |
|---|---|---|---|---|
| | P | A | C | S |
| FedCoda w/o BNC | 0.9369 | 0.9231 | 0.8790 | 0.9861 |
| FedCoda | 0.9903 | 0.9828 | 0.9848 | 0.9941 |

### 4.4 Sensitivity Analysis

### 4.4.1 Sensitivity Analysis of Hyperparameters in CFA

We analyze the sensitivity of FedCoda to the hyperparameters $\lambda_{FA}$ and $\lambda_{PC}$, which control the strengths of feature alignment and prototype calibration, respectively. We vary both coefficients in $\{0.5, 1.0, 2.0, 10.0\}$ and observe generalization accuracy on Digits with unseen domain *SVHN*. As shown in Figure 5, FedCoda maintains stable performance across a wide range of both hyperparameters. This behavior suggests that CFA does not rely on carefully tuned weights to be effective.

### 4.4.2 Sensitivity Analysis of Hyperparameters in Adaptive Gradient Clipping

We further examine the robustness of the proposed adaptive gradient clipping method with respect to its hyperparameters $\alpha$ and $\beta$, which control the clipping threshold scale and the smoothing factor of historical gradient norms, respectively. Specifically, we vary $\alpha$ in $\{1.0, 1.25, 1.5, 2.0\}$ and $\beta$ in $\{0.5, 0.9, 0.99, 0.999\}$, and evaluate performance on Digits with unseen domain *SVHN*. The results in Figure 6 show that FedCoda remains largely insensitive to the choice of both parameters. Across all hyperparameter settings, performance fluctuations are minor, indicating that the adaptive gradient clipping method is effective without requiring precise parameter tuning. This robustness is particularly important during the BN calibration stage, where gradient magnitudes may change abruptly. The ability to stabilize training across a wide range of $\alpha$ and $\beta$ values further demonstrates the practicality of the adaptive gradient clipping method.

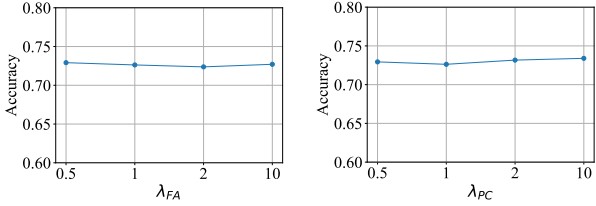
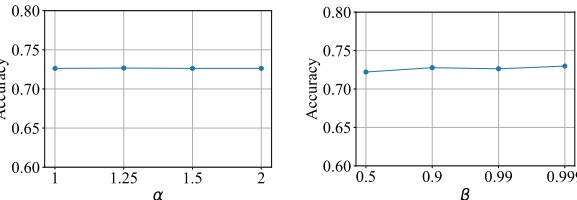

Figure 5: Sensitivity analysis of the hyperparameters $\lambda_{FA}$ (left) and $\lambda_{PC}$ (right) for calibrated feature alignment on Digits with unseen domain *SVHN*.

Figure 6: Sensitivity analysis of the hyperparameters $\alpha$ (left) and $\beta$ (right) for adaptive gradient clipping on Digits with unseen domain *SVHN*.

### 4.4.3 Analysis of Calibration Round

We study the impact of the calibration round, i.e., the communication round at which clients begin to use global BN statistics. Table 8 reports the Top-1 accuracy on PACS when the calibration stage starts at different rounds. Overall, the results indicate that FedCoda is robust to the choice of calibration round within a reasonable range. Starting calibration too early (e.g., at round 30) yields slightly inferior performance, as the global model and BN statistics have not yet sufficiently stabilized, leading to less reliable normalization. Conversely, delaying calibration excessively (e.g., at round 70) also degrades performance, since local models retain domain-specific BN statistics for too long, limiting the effectiveness of domain-invariant representation learning. The best performance is achieved when calibration begins around round 50. This setting enables a smooth transition from local to global normalization while preserving training stability. Based on these results, we adopt round 50 (i.e., the second half of the total training rounds) as the default calibration round in all experiments.

Table 8: Top-1 accuracy on PACS with different calibration rounds when begining to use global BN statistics.

| Rounds | PACS | | | |
|---|---|---|---|---|
| | P | A | C | S |
| 30 | 76.0±0.1 | 56.1±0.4 | 65.5±0.6 | 73.4±0.3 |
| 40 | 76.1±0.5 | 57.7±0.6 | 65.0±0.8 | 72.9±0.4 |
| 50 | 76.8±0.2 | 57.3±0.5 | 66.0±0.5 | 73.3±0.3 |
| 60 | 76.6±0.6 | 56.6±0.7 | 66.3±0.9 | 72.9±0.4 |
| 70 | 74.4±0.6 | 55.6±0.9 | 65.3±1.0 | 72.3±0.5 |

## 5 Conclusion

This work advances federated domain generalization by addressing feature inconsistency and domain imbalance under domain skew. We proposed FedCoda, a federated learning framework that integrates calibrated feature alignment and fair prototype aggregation to enable domain-invariant representation learning. Extensive experiments on multiple benchmarks demonstrate that FedCoda consistently outperforms relevant baselines and generalizes effectively to unseen domains. Experimental results show that calibrated feature alignment mitigates the adverse effects of mismatched BN statistics and inconsistent local prototypes, while prototype calibration further enhances inter-class separation and representation discriminability. On the server side, fair prototype aggregation prevents global prototypes from being biased toward dominant domains, leading to more stable and consistent optimization across clients. Together, these components form a cohesive solution for federated learning under domain skew. We hope that this work provides useful insights into feature alignment and fairness-aware aggregation, and inspires future research on domain generalization within the context of federated learning.

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

# A    Additional Experimental Results

## A.1    Generalization performance with Pretrained ResNet-18

Table 9 report the Top-1 accuracy on PACS with ResNet-18 pretrained on ImageNet. The experimental setting is the same as that in the main text. Experimental results show that FedCoda can still outperform other baselines under this setting. Besides, since the model is pretrained on ImageNet and its representation is better, the superiority of feature alignment is reduced in this setting.

Table 9: Top-1 accuracy (%) on PACS with ResNet-18 pretrained on ImageNet. All results are averaged over 3 runs (mean±std). The best results are highlighted in bold, while the second-best are underlined.

| Method | PACS | | | | |
|---|---|---|---|---|---|
| | P | A | C | S | Avg. |
| FedAvg | 84.9±0.5 | 67.5±2.0 | 71.2±0.7 | 77.5±0.7 | 75.3±0.7 |
| FedFM | 87.8±0.5 | 69.0±0.7 | 73.0±1.1 | 81.3±1.0 | 77.8±0.4 |
| FPL | 88.5±0.6 | 72.9±0.7 | 75.1±1.4 | 79.4±1.1 | 78.9±0.5 |
| FedHEAL | 77.4±0.7 | 62.3±0.6 | 71.2±0.8 | 79.7±0.4 | 72.6±0.4 |
| FedGMA | 83.7±0.4 | 68.3±0.9 | 68.3±1.1 | 78.8±0.7 | 74.8±0.4 |
| GA | 85.6±0.3 | 72.3±0.6 | 65.1±0.9 | 80.6±0.6 | 75.9±0.4 |
| FedIIR | 83.6±0.7 | 67.4±0.7 | 71.4±0.6 | 78.6±0.8 | 75.2±0.2 |
| FedSR | 88.5±0.9 | 69.8±2.0 | 73.4±2.0 | 77.4±1.7 | 77.2±0.9 |
| FixBN | 89.2±0.7 | 75.3±1.5 | 74.4±0.5 | **81.5±1.4** | 80.1±0.4 |
| FedCoda | **90.4±0.1** | **77.3±0.3** | **75.1±0.3** | 79.2±0.3 | **80.5±0.2** |

## A.2    Ablation of Core Components on DomainNet

To evaluate each component on a harder dataset, we conduct an ablation study on DomainNet. As shown in Table 10, both components contribute positively, and CFA is more helpful on DomainNet. Since the domain imbalance is milder for DomainNet, the benefit of FPA is more limited.

Table 10: Ablation study of core components of FedCoda on DomainNet. All results are averaged over 3 runs.

| FPA | CFA | Clipart | Infograph | Painting | Quickdraw | Real | Sketch | Avg. | Δ |
|---|---|---|---|---|---|---|---|---|---|
| | | 76.3±0.4 | 35.7±0.2 | 68.8±0.4 | 53.5±1.1 | 70.3±0.5 | 79.9±0.4 | 64.1±0.2 | - |
| ✓ | | 79.7±0.3 | 37.4±0.4 | 72.1±0.8 | 55.7±0.2 | 71.3±0.3 | 83.4±0.3 | 66.6±0.2 | 2.5 |
| | ✓ | 79.9±0.3 | 36.9±0.2 | 72.7±0.3 | 57.8±0.8 | 72.4±0.1 | 82.9±0.2 | 67.1±0.2 | 3.0 |
| ✓ | ✓ | 80.6±0.2 | 37.6±0.2 | 73.8±0.3 | 58.6±0.7 | 73.9±0.5 | 83.5±0.4 | **68.0±0.1** | **3.9** |

## A.3    Performance under Severe Imbalanced Domains

Apart from the client distribution setting in the main text, we further explore a more imbalanced setting, which can better evaluate the effect of FPA. Specifically, we initialize 13 clients for PACS. The number of clients varies across unseen domains: { P: 1, A: 1, C: 1, S: 10 } when the unseen domain is Photo; { P: 1, A: 1, C: 10, S: 1 } when the unseen domain is Art Painting; { P: 1, A: 10, C: 1, S: 1 } when the unseen domain is Cartoon; { P: 10, A: 1, C: 1, S: 1 } when the unseen domain is Sketch. Under this setting, we conduct an ablation study on FPA. As shown in Table 11, FPA is superior under this imbalanced setting, demonstrating that ensuring balanced global prototypes across domains can further improve the generalization performance.

Table 11: Ablation study of FPA on PACS with severe imbalanced distribution.

| Methods | PACS | | | | |
|---|---|---|---|---|---|
| | P | A | C | S | Avg. |
| without FPA | 76.0 | 53.8 | 59.7 | 71.5 | 65.3 |
| with FPA | 76.5 | 55.3 | 61.6 | 72.8 | 66.6 |

## A.4 Visualization of Feature Space

To qualitatively evaluate the effectiveness of feature alignment, we visualize the learned feature representations on PACS with Photo as the unseen target domain in Figure 7. Compared with competing methods, FedCoda produces more compact intra-class clusters and clearer inter-class separation, indicating more effective domain-invariant representation learning. Feature alignment methods (FedFM and FPL) partially improve class separability. However, their feature clusters remain less compact, suggesting that prototype inconsistency and aggregation bias still limit alignment quality. FixBN stabilizes representations by normalizing features with fixed BN statistics, but without explicit feature alignment, its decision boundaries remain ambiguous. In contrast, FedCoda yields the most structured feature space, with well-separated clusters and clearer decision boundaries. This improvement can be attributed to the combined effect of CFA and FPA. These mechanisms enable more reliable alignment between local features and global prototypes, resulting in improved generalization to unseen domains. To complement the qualitative analysis, we further evaluate representation quality using NMI. As reported in Table 12, FedCoda achieves the highest NMI score among all compared methods, confirming that it learns a more structured and informative feature space.

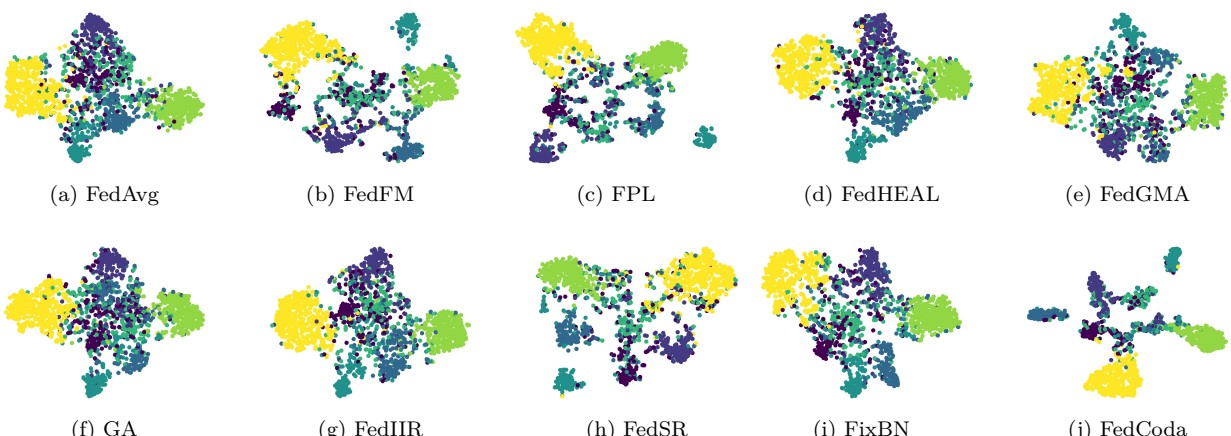

|  (a) FedAvg | (b) FedFM | (c) FPL | (d) FedHEAL | (e) FedGMA |
|---|---|---|---|---|
| (f) GA | (g) FedIIR | (h) FedSR | (i) FixBN | (j) FedCoda |

Figure 7: t-SNE visualization of representations on PACS dataset with unseen domain *Photo*. FedCoda facilitates more effective domain-invariant representation learning, as indicated by clearer decision boundaries on the unseen domain.

Table 12: NMI values of the feature spaces in Figure 7.

| Methods | FedAvg | FedFM | FPL | FedHEAL | FedGMA | GA | FedIIR | FedSR | FixBN | FedCoda |
|---|---|---|---|---|---|---|---|---|---|---|
| NMI ↑ | 0.5122 | 0.5631 | 0.5377 | 0.5123 | 0.4934 | 0.5043 | 0.5557 | 0.5402 | 0.5032 | **0.6452** |

## A.5 Performance over Communication Rounds

To examine the convergence behavior of different methods, we report the generalization accuracy over communication rounds in Figure 8. Overall, FedCoda consistently achieves higher accuracy and faster conver-

gence compared with competing approaches, indicating its effectiveness in stabilizing training under domain skew. On PACS, we observe a temporary drop in generalization accuracy at the beginning of the BN calibration stage (round 50). This transition reflects a shift from local BN statistics to global BN statistics, which introduces a mismatch between feature distributions and model parameters. In FedCoda, this effect is more pronounced than in FixBN because feature alignment and prototype-based losses are simultaneously optimized, making the model more sensitive to abrupt changes in feature normalization. In contrast, FixBN with gradient clipping does not involve prototype-based alignment, resulting in a smoother transition. Our adaptive gradient clipping mechanism mitigates this instability and enables rapid recovery after the transition. Note that original FixBN implementation does not include gradient clipping, and it fails to convergence after using global BN statistics (as shown in Table 6).

On PACS, we observe a temporary drop in generalization accuracy when the BN calibration stage begins (i.e., at round 50). This behavior is expected. At this stage, local models switch from domain-specific BN statistics to global BN statistics, leading to abrupt changes in feature normalization. The simultaneous introduction of feature alignment further amplifies discrepancies in optimization directions across clients, which can momentarily destabilize training and cause gradient explosion. Despite this shift, FedCoda rapidly recovers and converges to a higher performance level. This fast adaptation is largely attributed to the proposed adaptive gradient clipping strategy, which stabilizes optimization during the transition and allows local models to gradually align with global BN statistics. Overall, these results demonstrate that while BN calibration introduces short-term optimization challenges under severe domain shifts, FedCoda effectively mitigates this issue and achieves stable convergence.

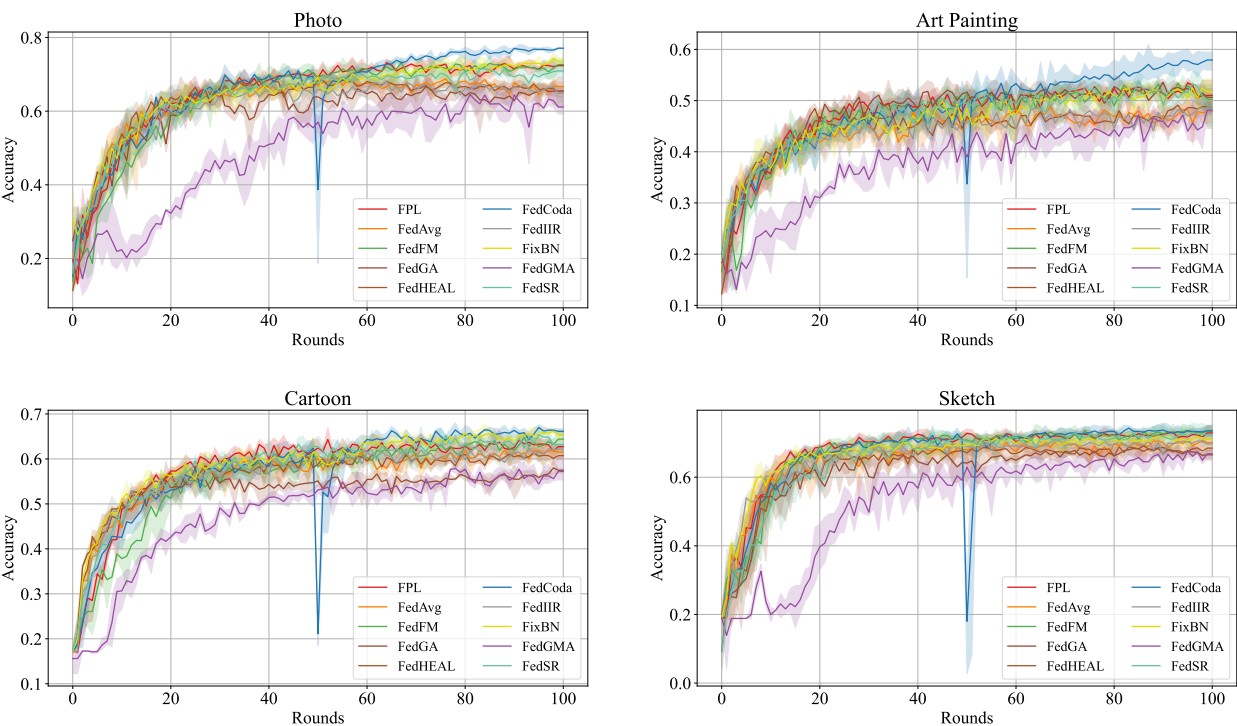

Figure 8: Generalization accuracy over communication rounds on PACS dataset. Due to new BN statistics, a temporary drop in generalization accuracy is observed when the calibration stage starts (i.e., at round 50). With the help of our adaptive gradient clipping, FedCoda rapidly adapts to the new BN statistics.

