# OpenReview forum: "Cross-Domain Feature Alignment for Federated Domain Generalization"
_TMLR — Rejected by TMLR_

### Review · Reviewer_mJ7L · 2026-03-07

**Summary Of Contributions:**

**Summary*
The paper proposes a way to do federated domain generalization by optimizing prototypes to help merge feature spaces across local clients.
The final method is a combination of adaptive BatchNorm statistics based on FixBN, contrastive learning objective, gradient clipping, and prototype separation.
The paper then provides a fair bit of experiments on pseudo-realistic Domain Generalization datasets of digits, PACS and DomainNet including ablation studies.

**Strengths**

- The paper has some interesting intuitions.

- The ablation studies were helpful.

**Weaknesses**

- The method is heuristic and complex, lacking proper justification and careful distinguishing from prior methods. Most components are very closely or exactly the same as prior works yet not explained carefully as such.
   - It starts with considering the effects of BN. Then, suggests to accomodate this by having some iterations with local BN statistics and later using global BN statistics.
   - But this leads to another problem of gradient explosion, so they suggest gradient clipping---but aren't there better methods for handling convergence issues like SCAFFOLD or similar methods? The gradient clipping method seems to be heuristic and is not justified with theory or good experiments compared to alternatives.
   - Then, a constrastive loss is developed to push prototypes away from each other but it is not directly compared to prior methods in contrastive learning or other prototype methods. This part does not seem to be novel but is likely very similar if not identical to prior contrastive learning losses. Again, this appears to be heuristic (Eq. 12).
   - The fair aggregation is another heuristic with almost no systematic justification for using this kind of weighting scheme.
      - The fair aggregation is weird. This will essentially find the center of the circle that goes through each of the local prototypes. This could be very far from average. Consider 3 points on a unit circle that are close in angle (45,50,55 degrees).  These points are very close to each other but the "prototype" your algorithm would find is the 0,0 point which is quite far from these 3 points.

- The paper does not compare using standard federated domain generalization benchmark [Bai et al., 2024], which carefully controls the experimental setting, implements multiple different methods, and shows challenging datasets. This would provide a fair comparison to existing Federated DG methods in a controlled and fair benchmark setting. Notice that FedAvg achieved more than 95% on PACS (though a slightly different setting). Thus, this calls into question the validity of the current results.  Also, many of the baselines used in this benchmarking work are not included in your work, for example, FedSR, FedDG, FedGMA, etc. Why are references to these methods missing?

Bai, R., Bagchi, S., & Inouye, D. I. Benchmarking Algorithms for Federated Domain Generalization. In The Twelfth International Conference on Learning Representations. 2024.

- Calibration stage is a hyperparameter. How do you know when to switch? It seems that you won't always know how many iterations T you should need to converge so it seems odd to choose T as a fixed parameter rather than some convergence criteria as would be more natural.

**Audience:**

Yes

**Audience Explanation:**

It is about federated learning and domain generalization which are clearly within the scope of TMLR.

**Broader Impact Concerns:**

No concerns about broader impacts.

**Claims And Evidence:**

No

**Claims Explanation:**

The methodological section lacks proper justification and contextualization within prior works. Many parts appear to be the paper's original idea but yet do not reference or explain how it is similar or different from closely related prior methods---e.g., the gradient clipping is only compared to vanilla gradient clipping but there are other clipping methods, similarly how is the contrastive loss functions similar or different from prior ones?

The empirical claims about outperforming prior methods is not substantiated. There is a strong benchmark for federated domain generalization [Bai et al., 2024] that showcases client heterogeneity and provides fair comparison to prior works. Without using this benchmarking paper, the validity of the empirical claims are questionable as this paper showed that FedAvg on PACS can do quite well even in heterogeneous settings.

**Requested Changes:**

- Justification for many of the methodological choices compared to the *best* alternatives and clear delineation about the novelty of your method compared to prior methods. This seems to be more like a systems paper that combines methods from other areas and/or heuristic methods that are actually similar to prior methods. But it is not framed that way. The contribution needs to be clear and the design choices well justified.

- Benchmarking using the framework from [Bai et al., 2024] in the same setting as the benchmark to enable a fair comparison to multiple DG methods (including FL ones that merely work on carefully converging). Importantly, this paper carefully creates an experimental setup and only allows each method a small hyperparameter tuning budget. Follow all the experimental protocols in that paper and compare on several of those datasets (including the harder ones like CelebA or IWildCam).

---

> ### Author Response · Authors · 2026-03-16
> **Response to Reviewer mJ7L (1/2)**
>
> Dear Reviewer mJ7L,
>
> Thank you for your detailed review and thoughtful comments. We believe there may be a few misunderstandings, and we are eager to address your concerns and clarify these potential misunderstandings.
>
> > W1: justification of methodology
>
> Thank you for the constructive feedback. We appreciate the reviewer’s comments regarding the justification of our design choices and the relationship between our method and prior work. In the revised manuscript, we have clarified these aspects and added additional analysis.
>
> 1. Clarifying the relationship to prior work.
>
> We have revised the manuscript to better distinguish our method from existing approaches. While prior work such as FixBN primarily focuses on improving optimization stability by fixing BN statistics in later training stages, our work investigates how mismatched BN statistics affect prototype-based feature alignment under domain skew. In prototype alignment methods, inconsistent BN statistics can produce divergent feature embeddings for the same class across clients, leading to **unreliable local prototypes and biased global prototypes**. We have added additional discussion in Sec. 3.3 to clarify this distinction and better position our contribution with respect to existing feature alignment methods.
>
> 2. Justification of adaptive gradient clipping.
>
> We agree that the motivation for adaptive gradient clipping should be explained more clearly. In the revision, we provide **both a clearer explanation and a theoretical analysis** of the proposed clipping mechanism, which prevents the update from becoming arbitrarily large when switching from local BN statistics to global BN statistics. Under standard smoothness and stochastic gradient assumptions, we prove that the clipped update satisfies a stability bound and remains a controlled approximation of the stochastic gradient.
>
> $F_k(\boldsymbol{w}^{s+1}) \le F_k(\boldsymbol{w}^s) - \eta \langle \nabla F_k(\boldsymbol{w}^s), \tilde{\boldsymbol{g}}^s \rangle + \frac{L\eta^2}{2}||\tilde{\boldsymbol{g}}^s||^2$
>
> $\mathbb{E}\big[||\tilde{\boldsymbol{g}}^s-\nabla F_k(\boldsymbol{w}^s)||^2 \big] \le 2\mathbb{E}\big[|| \tilde{\boldsymbol{g}}^s - \boldsymbol{g}^s||^2 \big] + 2\sigma^2$
>
> These results demonstrate that the clipping mechanism suppresses transient gradient spikes caused by BN mismatch while behaving similarly to standard SGD once optimization stabilizes.
>
> 3. Relation to optimization-based FL methods such as SCAFFOLD.
>
> We thank the reviewer for pointing out SCAFFOLD. SCAFFOLD addresses slow convergence in federated learning by introducing control variates to correct client drift and unify the optimization direction across clients. In contrast, our method focuses on improving representation consistency under domain skew. Feature alignment reduces intra-class variation and increases inter-class separation across clients, resulting in more consistent feature spaces.
>
> Overall, we have revised the manuscript to improve clarity regarding the motivation, novelty, and theoretical justification of our method, and to better position it relative to existing federated learning approaches.
>
> > W2: clarification of prototype calibration
>
> Thank you for the helpful comment. We agree that the motivation and positioning of the prototype calibration loss (Eq. 12) should be clarified with respect to existing contrastive learning approaches.
>
> Our intention is not to propose a new instance-level contrastive learning objective. Instead, the prototype calibration loss **operates at the class prototype level** and aims to encourage **a geometric structure among class representations**. Specifically, the loss promotes a simplex equiangular tight frame (ETF) structure, which maximizes angular separation among class prototypes and has been shown to improve class separability in representation learning.
>
> This differs from standard contrastive learning methods (e.g., InfoNCE-like loss used in FedFM), which rely on instance-level samples and prototypes. In contrast, our objective directly regularizes the geometry of class prototypes to improve the stability and consistency of prototype alignment under domain skew.
>
> We have revised the manuscript to clarify this distinction and to better position Eq. (12) as a geometric regularization for prototype representations rather than a novel contrastive learning loss.

---

> ### Author Response · Authors · 2026-03-16
> **Response to Reviewer mJ7L (2/2)**
>
> > W3: setting of fair prototype aggregation
>
> Thank you for the insightful geometric example. We agree that under perfectly symmetric settings (e.g., points lying on a circle), minimizing the variance of distances may produce a circumcenter-like solution that is not close to the prototype cluster. This behavior arises because the objective encourages balanced distances rather than minimizing the average distance. However, such symmetric settings correspond to degenerate geometric cases and are not representative of the domain-skew scenario considered in this work. In practice, local prototypes arise from heterogeneous feature distributions and typically form asymmetric clusters across domains. The purpose of the proposed aggregation objective is therefore to mitigate dominance from overrepresented domains rather than to estimate the arithmetic mean. We have revised Sec. 3.4 to clarify this geometric interpretation.
>
> > W4: robustness of switching point for the calibration stage
>
> Thank you for the helpful comment. We agree that the switching point for the calibration stage should not rely on knowing the exact convergence time in advance. In our method, the calibration stage is implemented as a simple training schedule rather than an assumption about the precise convergence point. We use T/2 as a practical default to allow sufficient early training with local BN statistics before introducing global BN statistics. Importantly, *the effectiveness of this schedule does not depend on accurately estimating convergence*. As shown in our sensitivity analysis (Table 5 in Section 4.4.3 of the original manuscript), FedCoda achieves similar performance across a wide range of switching rounds, indicating that the calibration stage is not a sensitive hyperparameter.
>
> We have revised Sec. 3.3.1 to clarify that the switching point should be interpreted as a coarse scheduling strategy rather than a strict convergence-based criterion.
>
> > W5: the experimental results on the benchmark (Bat et al., 2024)
>
> Thank you for pointing out the benchmark introduced by Bai et al. (2024). We agree that standardized benchmarks are valuable for evaluating federated domain generalization algorithms.
>
> - We evaluated FedCoda under their settings on PACS dataset. Using the benchmark setting with a ResNet-50 backbone pretrained on ImageNet, FedCoda achieves average accuracies of $0.9503\pm0.03$, $0.9449\pm0.04$, and $0.9317\pm0.02$ when $\lambda$ are 1, 0.1 and 0, respectively. These results demonstrate that FedCoda achieves comparable performance under their settings. We focused on PACS because it is a widely used domain generalization dataset and is used in this benchmark. Regarding other hard evaluation datasets, our experiments include DomainNet, which is widely recognized as a challenging benchmark for domain generalization due to its large scale and significant domain shift. Our original experiments follow the settings used in several recent federated DG works, including GA (Zhang et al., 2023), FPL (CVPR 2023), FedIIR (ICML 2023), and FedHEAL (CVPR 2024), which evaluate algorithms on datasets such as Digits, PACS, and DomainNet under similar settings. Nevertheless, we appreciate the reviewer’s suggestion, and clarified the relationship between our experimental setting and the benchmark (Bai et al., 2024) to avoid potential confusion when comparing the results.
>
> - The performance discrepancy for FedAvg on PACS arises from differences in experimental settings. In the benchmark (Bai et al., 2024), the model backbone is ResNet-50 pretrained on ImageNet, while our experiments use ResNet-18 trained from scratch. As explained in Sec. 4.1.3, we intentionally avoid pretrained models to better evaluate the effectiveness of the proposed feature alignment. Pretraining significantly improves baseline performance and can reduce the relative impact of representation learning, which explains why FedAvg achieves substantially higher accuracy in that benchmark.
>
> - We have added two baselines (FedSR and FedGMA) in our revised manuscript.
>
> - To ensure transparency and reproducibility, we provide the full source code for FedCoda and all compared baselines in an anonymous repository (https://anonymous.4open.science/r/FedCoda-0DE2/README.md). This allows reviewers to verify the experimental results.
>
>
> References:
>
> Ruipeng Zhang, Qinwei Xu, Jiangchao Yao, Ya Zhang, Qi Tian, and Yanfeng Wang. Federated domain generalization with generalization adjustment. CVPR 2023.
>
> Wenke Huang, Mang Ye, Zekun Shi, He Li, and Bo Du. Rethinking federated learning with domain shift: A prototype view. CVPR 2023.
>
> Yuhang Chen, Wenke Huang, and Mang Ye. Fair Federated Learning under Domain Skew with Local Consistency and Domain Diversity. CVPR 2024.
>
> Yaming Guo, Kai Guo, Xiaofeng Cao, Tieru Wu, and Yi Chang. Out-of-Distribution Generalization of Federated Learning via Implicit Invariant Relationships. ICML 2023.

---

> > ### Comment · Reviewer_mJ7L · 2026-04-02
> >
> > Thanks for the response. This partially addressed some of my concerns yet some of my concerns remain:
> >
> > 1. The method does not empirically beat FedAvg on PACS. PACS is a weird dataset as seen from the benchmarking paper. It is both harder and easier than other datasets. I believe to validate the empirical results, this needs to be tested on real-world rather than synthetic DG benchmark (as recommended by the benchmark paper in Remark 4.6). Without this and with the inability to beat FedAvg on PACS puts the empirical validity of the method still in question.
> >
> > 2. The aggregation method of trying to make the distances equal still seems like a poorly justified idea to me. This case is not an unusual case or corner case. If the points are not naturally spread out, it will place the point far from the points.
> >
> > 3. While the new justifications help, the paper would probably need a full re-review to determine if these significant modifications are sufficient to make all steps be properly justified.

---

> > > ### Author Response · Authors · 2026-04-03
> > >
> > > We thank the reviewer for the continued feedback. We believe there may be some misunderstandings.
> > >
> > > > Q1: The method does not empirically beat FedAvg on PACS.
> > >
> > > A: We would like to clarify that, as shown in Table 1, **FedCoda consistently outperforms FedAvg on PACS across all domains**, achieving an **average accuracy improvement (68.3% vs. 61.2%)**. FedCoda also outperforms FedAvg on **Digits and DomainNet**.
> > >
> > > We agree with the reviewer that PACS dataset is not enough to evaluate our method. To address this concern, we emphasize that, as shown in Table 2, **our method shows consistent improvements on DomainNet**, which is **larger-scale and more diverse**.
> > >
> > > Regarding the suggestion to evaluate on real-world datasets, we agree that this would further strengthen the work. Due to resource constraints, we focus on DG benchmarks widely adopted in the prior works published in the top-tier conferences, including GA (CVPR 2023), FPL (CVPR 2023), FedIIR (ICML 2023), and FedHEAL (CVPR 2024).
> > >
> > > > W2: Clarification of FPA.
> > >
> > > A: We acknowledge that FPA may not be optimal in all geometric settings. However, as shown in Table 3, we observe that **FPA consistently improves performance**. These results demonstrate that local prototypes arise from heterogeneous feature distributions and typically form asymmetric clusters across domains.
> > >
> > > > W3: Modification review.
> > >
> > > A: We thank the reviewer for this thoughtful comment. We would like to clarify that, while the revisions are substantial in presentation and analysis, **they do not alter the core methodology or experimental results of the paper**. Instead, the revisions focus on:
> > >
> > > - clarifying the relationship to prior work
> > >
> > > - providing theoretical analysis and ablation results for adaptive gradient clipping
> > >
> > > - clarifying the details of core components
> > >
> > > In particular, the key components of our method (CFA and FPA), the training procedure, and the reported results **remain unchanged**. The additional experiments and analyses are designed to **make the underlying mechanisms more explicit and better supported**, rather than introducing new assumptions or modifying the algorithm. We hope that these revisions make the paper easier to evaluate and address the concerns raised in the initial review.

---

> > > > ### Comment · Reviewer_mJ7L · 2026-04-03
> > > >
> > > > Thank you for your response. I will take these under consideration for my final recommendation.
> > > >
> > > > Q1 - I was meaning performance on PACS for the Federated DG benchmark across lambda values, not the original results in your paper. The Federated DG benchmark is a more fair setting for comparison as FedAvg was optimized independently of the other methods. Thus, under the benchmark settings, there is (as of yet) no evidence that it is better than FedAvg.

---

> > > > > ### Author Response · Authors · 2026-04-14
> > > > >
> > > > > Thanks for raising this point regarding evaluation under the Federated DG benchmark. We acknowledge that our method does not achieve superior performance under this benchmark. This suggests that handling the combined challenges of domain skew and client heterogeneity remains non-trivial, and that improvements targeting domain skew alone may not directly translate to gains in more complex federated settings.
> > > > >
> > > > > We clarify that the primary goal of this work is to isolate and study feature alignment under domain skew. Therefore, **we follow prior works that only consider domain skew for clearer analysis**. Under this setting, our method demonstrates consistent improvements, supporting the effectiveness of our CFA and FPA.
> > > > >
> > > > > At the same time, the Federated DG benchmark represents a more general and challenging scenario. We view this as an important direction for future work, where extending feature alignment to better handle additional heterogeneity could further improve performance.

---

### Review · Reviewer_zTtf · 2026-03-21

**Summary Of Contributions:**

This paper studies federated domain generalization under domain skew and proposes **FedCoda**, which combines two components: **Calibrated Feature Alignment (CFA)** and **Fair Prototype Aggregation (FPA)**. CFA uses a later-stage switch from local BN statistics to global BN statistics, adaptive gradient clipping, and a prototype calibration loss intended to improve prototype consistency and inter-class separation. FPA aggregates local prototypes by optimizing for more balanced global prototypes under domain skew. The paper evaluates the method on Digits, PACS, and a reduced DomainNet benchmark and reports consistent gains over the listed baselines.

The main strengths are that the problem is relevant, the high-level decomposition into prototype calibration and fair aggregation is intuitive, and the empirical results are encouraging, especially on PACS. The main weaknesses are that the novelty relative to **FixBN** is not positioned clearly enough, several central claims are stronger than the evidence provided, important baselines and ablations are missing, and some implementation details and explanations are inconsistent.

**Additional Comments:**

Overall, I think the paper is promising, but the current version overstates what has been established. The most important revision would be to position the method more honestly as an extension of FixBN-style BN calibration plus prototype aggregation, and then provide the ablations and clarifications needed to show which parts are actually responsible for the gains.

**Audience:**

Yes

**Audience Explanation:**

Yes. The topic is clearly relevant to TMLR’s audience. Researchers working on federated learning, domain generalization, representation learning, and prototype-based FL would likely be interested in the problem and in the reported empirical gains. Even though I do not think the current version fully supports all of its claims, the problem setting is important and the proposed direction is of clear interest to the community.

**Broader Impact Concerns:**

I do not see a major ethical concern beyond the privacy wording. The main issue is that the manuscript currently makes stronger privacy claims than the evidence justifies. In a federated learning paper, statements about privacy should be particularly careful and should not be presented as substantive guarantees without direct support.

**Claims And Evidence:**

No

**Claims Explanation:**

The empirical results are promising, but the evidence does not yet fully support several of the paper’s central claims.

First, the BN-calibration part of CFA is very close in spirit to **FixBN**, and the paper itself says it is inspired by FixBN. However, the novelty relative to that baseline is not isolated clearly enough. The current experiments do not cleanly disentangle how much of the gain comes from the FixBN-style BN calibration, how much comes from adaptive clipping, and how much comes from FPA.

Second, several mechanistic claims are asserted rather than directly demonstrated. The paper claims that mismatched BN statistics harm feature alignment and prototype quality, and that applying global BN statistics under domain skew can lead to gradient explosion. However, there is no direct experiment showing prototype inconsistency before and after BN calibration, no direct measurement of gradient norms or clipping behavior, and no theory that convincingly establishes these claims in the specific setting studied here.

Third, the paper’s discussion of **FixBN** and the round-50 behavior in Fig. 7 is currently not convincing. The original FixBN method is itself a **two-stage training procedure** that switches to frozen global BN statistics around the midpoint of training. The submission also states that FixBN begins BN calibration at round 50. Therefore, if the drop in FedCoda at round 50 is attributed mainly to switching from local BN statistics to global BN statistics, the paper should explain why a similar transition effect is not visible for FixBN. This does not necessarily imply that the implementation is wrong, but it does mean that the current explanation is incomplete. In addition, the paper says the round-50 drop is partly caused by the “simultaneous introduction of feature alignment,” but elsewhere it states that feature alignment is absent only in the first round. These statements are difficult to reconcile.

Fourth, the fairness and privacy claims are too strong relative to the evidence. FPA is described as ensuring fairness across domains, but the paper does not provide fairness-oriented metrics such as worst-domain accuracy or variance across domains. Likewise, the claim that the approach preserves category-wise privacy is too strong given that class-wise prototypes are still communicated.

Fifth, there are methodological inconsistencies that affect clarity and reproducibility. The formal setup in Section 3.1 assumes consistent $P(Y \mid X)$, but the experimental setup later says that some benchmarks contain both domain skew and label skew. Algorithm 1 and the surrounding text are also not fully consistent about local versus global BN statistics, when clipping is applied, and when feature alignment is active.

Overall, the paper presents a plausible and interesting method, but some of its strongest claims are not yet supported by sufficiently direct or convincing evidence.

**Requested Changes:**

### Critical to securing acceptance

1. **Reposition the method more clearly relative to FixBN.**
   The paper should make it explicit that the BN-calibration part of CFA is an adaptation/extension of **FixBN-style two-stage BN calibration** for prototype-based federated domain generalization, rather than leaving the novelty boundary unclear.

2. **Add stronger ablations around the FixBN base.**
   The current experiments do not isolate the source of the improvement well enough. The paper should compare at least:
   - FixBN,
   - FixBN + standard/static clipping,
   - FixBN + FPA,
   - FixBN + dynamic clipping,
   - full FedCoda.

3. **Clarify the FixBN implementation and the round-50 behavior in Fig. 7.**
   Since the original FixBN is also a two-stage method with a midpoint BN switch, and the paper states that FixBN calibration begins at round 50, the authors should explain:
   - how FixBN is implemented in this paper,
   - whether it uses the original two-stage procedure faithfully,
   - whether the reported FixBN includes additional clipping not present in the original method,
   - why the round-50 drop is pronounced for FedCoda but not visibly similar for FixBN,
   - whether the drop is due mainly to BN switching, feature alignment, or the interaction between the two.

4. **Substantiate or soften the claim that domain skew can cause gradient explosion during BN calibration.**
   This is currently asserted without direct support. The authors should provide either a targeted experiment, a convincing reference, or a clearer theoretical argument.

5. **Provide direct evidence for the claimed BN-mismatch mechanism.**
   If the paper claims to investigate the adverse effect of mismatched BN statistics on feature alignment, then it should directly measure that effect, for example through prototype divergence, alignment quality, gradient norms, or accuracy before and after the BN switch.

6. **Tone down or support the fairness and privacy claims.**
   In particular, claims about fairness across domains and category-wise privacy preservation should either be backed by direct evidence or restated more conservatively.

7. **Resolve inconsistencies in the method description.**
   The paper should clarify:
   - the role of the 80/20 split under leave-one-domain-out evaluation,
   - whether clipping is used only during calibration or throughout training,
   - when feature alignment is actually introduced,


### Would strengthen the work

1. Add diagnostics for the clipping mechanism, such as average clipping coefficient, fraction of clipped parameters, or gradient norm over training rounds.

2. Extend the sensitivity analysis to wider parameter ranges so that it shows both robustness and failure regions.

3. Add communication/computation overhead analysis.

4. Improve the title and the framing question in the introduction so that they better reflect the actual contribution.

5. Perform another careful pass for typos, wording, and consistency.

---

> ### Author Response · Authors · 2026-03-24
> **Response to Reviewer zTtf (1/2)**
>
> Dear Reviewer zTtf,
>
> Thank you for valuable comments and acknowledgement of our approach. We are eager to address your concerns and clarify some potential misunderstandings.
>
> > W1 & C1: Reposition the method.
>
> We thank the reviewer for this important observation. We agree that the original submission did not clearly position our method relative to FixBN, nor did it sufficiently isolate the contributions of each component.
>
> We have revised the Introduction and Methodology section (Sec. 3.3) to explicitly clarify that CFA builds upon the two-stage BN calibration strategy of FixBN, and that our contributions are extensions tailored to prototype-based federated domain generalization under domain skew. Specifically, we now clearly distinguish the roles of each component:
>
> - FixBN: introduces two-stage training with global BN statistics in later rounds.
> - Our extension (CFA): (1) Adaptive gradient clipping, which stabilizes training under large discrepancies between local and global BN statistics; (2) Prototype calibration loss, which improves inter-class separation and prototype consistency.
> - FPA (orthogonal component): addresses bias in global prototype aggregation under domain imbalance.
>
> In the revised manuscript, we describe CFA as a BN calibration mechanism augmented for prototype alignment.
>
> > W2, C2, C4 & C5: Add stronger ablations around the FixBN and provide direct evidence for the claimed BN-mismatch mechanism.
>
> To directly address the reviewer's concern, we have added a new ablation study (Table 6 in Sec. 4.3.3) that demonstrate the contribution of adaptive gradient clipping.
>
> | Methods                              | P           | A           | C           | S           |
> |--------------------------------------|-------------|-------------|-------------|-------------|
> | FixBN w/o gradient clipping          | 11.3 ± 0.0  | 18.5 ± 0.0  | 16.6 ± 0.0  | 19.6 ± 0.0  |
> | FixBN with fixed clipping            | 73.0 ± 0.6  | 51.4 ± 0.9  | 65.4 ± 0.6  | 71.2 ± 0.4  |
> | FixBN with adaptive clipping         | 74.7 ± 0.5  | 53.8 ± 0.9  | 64.5 ± 0.6  | 71.7 ± 0.4  |
> | FedCoda                              | **76.8 ± 0.2** | **57.3 ± 0.5** | **66.0 ± 0.5** | **73.3 ± 0.3** |
>
> The results show that vallina FixBN (i.e., FixBN without gradient clipping) fails to converge when directly applying global BN statistics under domain skew. Besides, our adaptive gradient clipping can also improve FixBN, which further enhances the contribution of our adaptive gradient clipping.
>
> To provide the direct evidence for the claimed BN-mismatch mechanism, we added new results measuring the gradient norms over training rounds. Figure 4 in Sec. 4.3.3 shows that at the BN switching point (round 50), gradient norms increase sharply under domain skew and our adaptive gradient clipping effectively stabilizes gradient magnitudes. This confirms that BN switching introduces optimization instability, especially under domain skew, and motivates our adaptive clipping mechanism.
>
> > W3 & C3: Clarify the FixBN implementation and the round-50 behavior.
>
> We thank the reviewer for this careful analysis. We agree that the original manuscript did not sufficiently clarify the implementation details of FixBN and the behavior observed at the BN switching point. We have revised the paper to address these issues in detail.
>
> We clarify that our implementation of FixBN follows the original two-stage training procedure. However, for training stability under domain skew, we additionally apply standard gradient clipping (threshold = 10) in our FixBN implementation. This modification is explicitly stated in Sec.4.1.2 in the original manuscript. In the revised manuscript, we emphasize that this addition does not change the core FixBN mechanism and the reported FixBN includes additional clipping not present in the original method.
>
> For the reason why the round-50 drop appears in FedCoda but not in FixBN, we have revised Sec. 4.6 to clarify that the observed drop in FedCoda arises from the interaction between BN switching and prototype-based feature alignment, rather than BN switching alone. Specifically, in FixBN, switching BN statistics affects only feature normalization, and the training objective remains unchanged. In FedCoda, the model simultaneously optimizes classification loss, feature alignment loss and prototype calibration loss. At the BN switching point, this leads to a more pronounced transient instability in FedCoda compared to FixBN.

---

> ### Author Response · Authors · 2026-03-24
> **Response to Reviewer zTtf (2/2)**
>
> > W4 & C6: Tone down or support the fairness and privacy claims.
>
> We thank the reviewer for pointing out that our fairness and privacy claims were overstated relative to the provided evidence. We agree with this assessment and have revised the claims to ensure they are accurate and well-supported. Specifically, We emphasize that the design of FPA explicitly minimizes variance in distances between global and local prototypes, which encourages balanced representation across domains. Besides, we have clarified that FPA does not rely on sharing explicit label distribution statistics.
>
> > W5 & C7: Resolve inconsistencies in the method description.
>
> We thank the reviewer for carefully identifying these issues. We agree that several parts of the original manuscript could be clarified, and we have revised the manuscript to ensure consistency and precise definitions.
>
> We would like to clarify that our experimental setting is consistent with the problem formulation. Specifically, the conditional distribution $P(\mathcal{Y}|\mathcal{X})$ remains consistent across clients. The heterogeneity arises from feature skew (variation in $P(\mathcal{X})$) and label skew (variation in $P(\mathcal{Y})$).
>
> For the role of the 80/20 split under leave-one-domain-out evaluation, we have clarified that the samples in each training domain are split into 80\% for training and 20\% for testing.
>
> For the usage of BN calibration, we have revised Algorithm 1 to clearly specify: (1) clients use local BN statistics when $t < T/2$; (2) clients switch to global BN statistics when $t >= T/2$.
>
> For the usage of adaptive gradient clipping, we have revised Algorithm 1 to clearly specify that it is used after using global BN statistics.
>
> For the timing of feature alignment, we have clearly stated that feature alignment is enabled from the second round, once the global prototypes available.
>
> > Would strengthen the work
>
> We thank the reviewer for these constructive suggestions. We agree that these additions and refinements would improve the clarity and completeness of the paper.
>
> 1. We agree that additional diagnostics would make the role of adaptive gradient clipping more transparent. In the revised manuscript, we have added the results of gradient norms over training rounds. These additions strengthen our empirical support for the stabilization effect of the proposed clipping strategy.
>
> 2. The goal of our sensitivity analysis is to evaluate the practical robustness of the method within a reasonable range of hyperparameters. The current ranges (e.g., $\\lambda_{FA}, \\lambda_{PC} \in \\{0.5, 1.0, 2.0, 10.0\\} $, and similar ranges for clipping parameters) already covers both moderate and relatively large values.
>
> 3. We agree that overhead analysis would improve the practical characterization of our method. In the revised manuscript, we add a discussion of communication overhead. In particular, we clarify that our method only adds transmission of class-wise prototypes, while FPL requires transmitting multiple clustered prototypes.
>
> 4. In the revised manuscript, we improve the statement in the Introduction to better illustrate the scope and contribution of this paper.
>
> 5. We appreciate this comment and agree that the manuscript would benefit from a careful check. In the revision, we perform a thorough pass to correct typos and resolve inconsistencies across the method description, algorithm, and experiments.

---

> > ### Comment · Reviewer_zTtf · 2026-04-09
> >
> > The rebuttal and revision address **several of the clarity and evidence issues** I raised, especially the **positioning relative to FixBN**, the **implementation details of the FixBN** baseline, and the **optimization-instability claim around BN switching**. However, some key concerns remain only partially resolved, particularly the incomplete ablation structure around the FixBN base, the lack of direct evidence for the claimed BN-mismatch effect on prototype quality, and the still limited empirical support for the fairness claims.
> >
> > * Please add an ablation for FixBN + FPA so that the contribution of the fair prototype aggregation module can be isolated from the BN-calibration and clipping components.
> >
> > * Please provide a direct diagnostic of prototype inconsistency before and after BN calibration (for example, prototype divergence, prototype variance across clients, or alignment quality), so that the claimed BN-mismatch mechanism is supported more directly. Otherwise, please soften the claim to optimization instability.

---

> > > ### Author Response · Authors · 2026-04-14
> > >
> > > We thank the reviewer for the continued feedback. Here we address the remaining concerns.
> > >
> > > > Please add an ablation for FixBN + FPA so that the contribution of the fair prototype aggregation module can be isolated from the BN-calibration and clipping components.
> > >
> > > A: Thanks for your comment. Since FixBN does not incorporate feature alignment, FixBN cannot be integrated with FPA. To further evaluate the effect of fair prototype aggregation, we have conducted an additional ablation study on a scenario where the client distribution is more imbalanced across domains (as suggested by the Reviewer nnNV).
> > >
> > > | Methods      | P    | A    | C    | S    | Avg. |
> > > |--------------|------|------|------|------|------|
> > > | without FPA  | 76.0 | 53.8 | 59.7 | 71.5 | 65.3 |
> > > | with FPA     | 76.5 | 55.3 | 61.6 | 72.8 | 66.6 |
> > >
> > > The experimental results show that FPA achieves better generalization performance under such imbalanced scenario, which demonstrates that ensuring balanced global prototypes across domains can further improve the generalization performance.
> > >
> > > > Please provide a direct diagnostic of prototype inconsistency before and after BN calibration (for example, prototype divergence, prototype variance across clients, or alignment quality), so that the claimed BN-mismatch mechanism is supported more directly. Otherwise, please soften the claim to optimization instability.
> > >
> > > A: Thanks for this insightful comment. To better understand the role of BN calibration, we measure the cosine similarity of local prototypes across clients.
> > >
> > > | Methods           | P      | A      | C      | S      |
> > > |-------------------|--------|--------|--------|--------|
> > > | FedCoda w/o BNC   | 0.9369 | 0.9231 | 0.8790 | 0.9861 |
> > > | FedCoda           | 0.9903 | 0.9828 | 0.9848 | 0.9941 |
> > >
> > > The experimental results demonstrate that BN Calibration (BNC) can enhance the prototype consistency across clients. In particular, this mismatch is more severe when the training domains contain Sketch (i.e., when the unseen domains are Photo, Art Painting and Cartoon), whose style is much different from other domains. However, BN calibration can effectively mitigate the mismatched BN statistics under this setting, thereby improving the consistency of local prototypes.

---

### Review · Reviewer_nnNV · 2026-04-05

**Summary Of Contributions:**

This paper introduces FedCoda, an FL method for domain generalization under skew. The authors note that the feature inconsistencies lead to a mismatched batch normalization statistics and weak class separation, and global prototype aggregation becomes biased toward overrepresented domains.

FedCoda addresses this with two components. First, Calibrated Feature Alignment (CFA) improves prototype quality. It switches from local BN statistics to global BN statistics in the later stage of training and uses adaptive gradient clipping to avoid instability during that switch. It also adds a prototype calibration loss that encourages better inter-class separation using an ETF-like geometric structure. Second, Fair Prototype Aggregation (FPA) computes global prototypes by minimizing the variance of distances from the global prototype to local prototypes, rather than simply averaging them.

**Additional Comments:**

This is a nice paper, so I recommend accept

**Audience:**

Yes

**Audience Explanation:**

This solves an important problem in FL.

**Claims And Evidence:**

Yes

**Claims Explanation:**

This paper has strong empirical performance and good ablations.

**Requested Changes:**

1.	The paper argues that prototype-based FL methods fail because of both BN mismatch and aggregation bias. Can the authors provide a cleaner decomposition of how much each factor contributes independently, beyond the current ablations? For example, is prototype inconsistency mostly a BN issue, or does weak inter-class separation play an equally large role?

2.	The paper defines fairness in prototype aggregation through minimizing the variance of distances from local prototypes to the global prototype. Why is this the right notion of fairness for federated domain generalization? Are there cases where equidistance could overemphasize noisy or outlier domains?

3.	The method combines BN calibration, adaptive clipping, feature alignment, and prototype calibration. Which of these components is most essential for the gains on the hardest benchmark, DomainNet? The current ablations focus more on Digits and PACS.

4. The paper does not use ImageNet-pretrained backbones, unlike some prior works. This is understandable, but could the authors also include experiments with pretrained backbones so the comparison to methods such as GA and FedIIR is more directly comparable to prior literature?

 5. The client distribution is intentionally imbalanced across domains. Have the authors tested more extreme imbalance patterns, where one domain dominates even more strongly? That seems especially relevant to the motivation for FPA.

---

> ### Author Response · Authors · 2026-04-14
> **Response to Reviewer nnNV (1/2)**
>
> Dear Reviewer nnNv,
>
> Thank you for your acknowledgement and recommendation. Your constructive suggestions have greatly improved our final manuscript. Here we address your concerns.
>
> > Q1: The paper argues that prototype-based FL methods fail because of both BN mismatch and aggregation bias. Can the authors provide a cleaner decomposition of how much each factor contributes independently, beyond the current ablations? For example, is prototype inconsistency mostly a BN issue, or does weak inter-class separation play an equally large role?
>
> A: We thank the reviewer for this insightful suggestion. We agree that a cleaner decomposition would strengthen the paper.
>
> In the revised manuscript, we conduct an ablation study on two components of CFA, i.e., BN Calibration (BNC) and Prototype Calibration (PC).
>
> | Method              | M    | MM   | SV   | SY   | Avg. | P    | A    | C    | S    | Avg. |
> |---------------------|------|------|------|------|------|------|------|------|------|------|
> | FedCoda w/o BNC     | 94.7 | 57.9 | 68.8 | 93.7 | 78.8 | 72.7 | 53.2 | 64.1 | 72.9 | 65.7 |
> | FedCoda w/o PC      | 96.4 | 59.6 | 72.1 | 93.7 | 80.5 | 76.2 | 56.2 | 64.9 | 72.2 | 67.4 |
> | FedCoda             | 96.5 | 61.2 | 72.6 | 94.2 | **81.1** | 76.8 | 57.3 | 66.0 | 73.3 | **68.3** |
>
> Ablation results show that BNC and PC consistently improve generalization performance on unseen domains, and BNC plays a more essential role. We have added this ablation study in the revised manuscript (Table 4 in Sec. 4.3.1).
>
> > Q2: The paper defines fairness in prototype aggregation through minimizing the variance of distances from local prototypes to the global prototype. Why is this the right notion of fairness for federated domain generalization? Are there cases where equidistance could overemphasize noisy or outlier domains?
>
> A: We thank the reviewer for this thoughtful question. We agree that the notion of fairness adopted in FPA requires clearer justification and careful interpretation.
>
> Our intention is not to claim that minimizing distance variance is the unique or universally optimal definition of fairness, but rather to provide a practical and geometry-based surrogate for mitigating aggregation bias under domain imbalance. Minimizing the variance of distances serves as a way to encourage the global prototype to be comparably representative across domains. In this sense, our notion of fairness is better interpreted as balancing representational influence across domains, rather than enforcing strict equality.
>
> We agree with the reviewer that enforcing fair aggregation could overemphasize noisy prototypes. However, this can be mitigated as the training proceeds. Similarly, the prototypes at the early stage of FL can be noisy, but these prototypes tend to be representative as the model converges. Therefore, the influence of those noisy prototypes can be mitigated as the model converges.
>
> > Q3: The method combines BN calibration, adaptive clipping, feature alignment, and prototype calibration. Which of these components is most essential for the gains on the hardest benchmark, DomainNet? The current ablations focus more on Digits and PACS.
>
> A: Thanks for your constructive comment. We have added the ablation study on DomainNet.
>
> | FPA | CFA | Clipart | Infograph | Painting | Quickdraw | Real | Sketch | Avg. | $\\Delta$ |
> |-----|-----|----------|------------|----------|------------|------|--------|------|----|
> |     |     | 76.3±0.4 | 35.7±0.2  | 68.8±0.4 | 53.5±1.1  | 70.3±0.5 | 79.9±0.4 | 64.1±0.2 | - |
> | ✓   |     | 79.7±0.3 | 37.4±0.4  | 72.1±0.8 | 55.7±0.2  | 71.3±0.3 | 83.4±0.3 | 66.6±0.2 | 2.5 |
> |     | ✓   | 79.9±0.3 | 36.9±0.2  | 72.7±0.3 | 57.8±0.8  | 72.4±0.1 | 82.9±0.2 | 67.1±0.2 | 3.0 |
> | ✓   | ✓   | 80.6±0.2 | 37.6±0.2  | 73.8±0.3 | 58.6±0.7  | 73.9±0.5 | 83.5±0.4 | 68.0±0.1 | 3.9 |
>
> The experimental results show similar patterns. Both components contribute positively, and their combination yields the largest performance gains across all evaluated settings. Besides, CFA is more essential on DomainNet. We have added this ablation study in the revised manuscript (Table 10 in Appendix A.2).

---

> ### Author Response · Authors · 2026-04-14
> **Response to Reviewer nnNV (2/2)**
>
> > Q4: The paper does not use ImageNet-pretrained backbones, unlike some prior works. This is understandable, but could the authors also include experiments with pretrained backbones so the comparison to methods such as GA and FedIIR is more directly comparable to prior literature?
>
> A: Thanks for your insightful suggestion. As shown in Table 9 in Appendix A.1, we have added the experimental results with ResNet-18 pretrained on ImageNet.
>
> | Method | Photo | Art Painting | Cartoon | Sketch | Avg. |
> |---|---|---|---|---|---|
> | FedAvg | 84.9±0.5 | 67.5±2.0 | 71.2±0.7 | 77.5±0.7 | 75.3±0.7 |
> | FedFM | 87.8±0.5 | 69.0±0.7 | 73.0±1.1 | 81.3±1.0 | 77.8±0.4 |
> | FPL | 88.5±0.6 | 72.9±0.7 | 75.1±1.4 | 79.4±1.1 | 78.9±0.5 |
> | FedHEAL | 77.4±0.7 | 62.3±0.6 | 71.2±0.8 | 79.7±0.4 | 72.6±0.4 |
> | FedGMA | 83.7±0.4 | 68.3±0.9 | 68.3±1.1 | 78.8±0.7 | 74.8±0.4 |
> | GA | 85.6±0.3 | 72.3±0.6 | 65.1±0.9 | 80.6±0.6 | 75.9±0.4 |
> | FedIIR | 83.6±0.7 | 67.4±0.7 | 71.4±0.6 | 78.6±0.8 | 75.2±0.2 |
> | FedSR | 88.5±0.9 | 69.8±2.0 | 73.4±2.0 | 77.4±1.7 | 77.2±0.9 |
> | FixBN | 89.2±0.7 | 75.3±1.5 | 74.4±0.5 | **81.5±1.4** | 80.1±0.4 |
> | FedCoda | **90.4±0.1** | **77.3±0.3** | **75.1±0.3** | 79.2±0.3 | **80.5±0.2** |
>
> Experimental results show that FedCoda can still outperform other baselines under this setting. Besides, since the model is pretrained on ImageNet and its representation is better, the superiority of feature alignment is reduced in this setting.
>
> > Q5: The client distribution is intentionally imbalanced across domains. Have the authors tested more extreme imbalance patterns, where one domain dominates even more strongly? That seems especially relevant to the motivation for FPA.
>
> A: Thanks for your constructive comment. We further explore a more imbalanced setting, which can better evaluate the effect of FPA. Specifically, we initialize 13 clients for PACS. The number of clients varies across unseen domains: { P: 1, A: 1, C: 1, S: 10 } when the unseen domain is Photo; { P: 1, A: 1, C: 10, S: 1 } when the unseen domain is Art Painting; { P: 1, A: 10, C: 1, S: 1 } when the unseen domain is Cartoon; { P: 10, A: 1, C: 1, S: 1 } when the unseen domain is Sketch.
>
> | Methods      | P    | A    | C    | S    | Avg. |
> |--------------|------|------|------|------|------|
> | without FPA  | 76.0 | 53.8 | 59.7 | 71.5 | 65.3 |
> | with FPA     | 76.5 | 55.3 | 61.6 | 72.8 | 66.6 |
>
> The experimental results show that FPA achieves better generalization performance under such imbalanced scenario, which demonstrates that ensuring balanced global prototypes across domains can further improve the generalization performance.

---

### Decision · Action_Editor_PzMj · 2026-05-18

**Recommendation:** Reject

**Audience:**

Yes

**Audience Explanation:**

interesting and important problem

**Claims And Evidence:**

No

**Claims Explanation:**

The manuscript addresses the critical challenge of domain skew in federated domain generalization by introducing FedCoda, a framework consisting of Calibrated Feature Alignment (CFA) and Fair Prototype Aggregation (FPA). The reviewers generally agree that the targeted problem is highly relevant to the machine learning community and that the high-level decomposition of the method is intuitive. Throughout the interactive review process, the authors successfully engaged in substantial revisions that significantly clarified the paper's positioning. Furthermore, the inclusion of targeted ablation studies, empirical evidence regarding batch-normalization (BN) mismatch, and diagnostics illustrating how adaptive gradient clipping suppresses optimization instability around the BN switching point effectively resolved several core technical concerns raised by the panel. Consequently, two of the three reviewers recommend acceptance, noting the solid empirical performance.

Despite these empirical contributions, several critical reservations remain regarding the fundamental justification of the method. The concerns appear to be at best partially addressed, casting doubt on whether its primary claims are fully supported. Chief among these is the lack of a cohesive narrative; the framework relies on multiple complex components that feel more like disconnected heuristics rather than well-justified, systematic design choices. Furthermore, when evaluated under a rigorously controlled Federated DG benchmark using pre-trained backbones, the proposed method fails to clearly outperform standard FedAvg. While the authors transparently acknowledge that their work isolates feature alignment under domain skew rather than generic client heterogeneity, this discrepancy means there is insufficient positive evidence that the structural innovations translate to meaningful empirical improvements in more realistic scenarios. Consequently, the manuscript still reads as a collection of experimental results rather than a conceptually tight study, and these significant structural ambiguities and justification gaps must be resolved before the paper can be considered ready for publication.